

# An ion-neutral model to investigate chemical ionization mass spectrometry analysis of atmospheric molecules - application to a mixed reagent ion system for hydroperoxides and organic acids

Brian G. Heikes[1], Victoria Treadaway[1], Ashley S. McNeill[1,2], Indira K. C. Silwal[3,4] and Daniel W. O'Sullivan[3]

[1]Graduate School of Oceanography, University of Rhode Island, Narragansett, RI, 02882, USA
[2]Department of Chemistry, The University of Alabama, Tuscaloosa, AL, 35401, USA
[3]Chemistry Department, United States Naval Academy, Annapolis, MD, 21402, USA
[4]Department of Chemistry, University of Maine, Orono, 04469, USA

*Correspondence to*: Brian Heikes (bheikes@uri.edu)

**Abstract.** An ion-neutral chemical kinetic model is described and used to simulate the negative-ion chemistry occurring within a mixed-reagent ion chemical ionization mass spectrometer (CIMS). The model objective was the establishment of a theoretical basis to understand ambient pressure (variable sample flow and reagent ion carrier gas flow rates), water vapor, ozone and oxides of nitrogen effects on ion-cluster sensitivities for hydrogen peroxide ($H_2O_2$), methyl peroxide ($CH_3OOH$), formic acid ($HFo$) and acetic acid ($HAc$). The model development started with established atmospheric ion chemistry mechanisms, thermodynamic data and reaction rate coefficients. The chemical mechanism was augmented with additional reactions and their reaction rate coefficients specific to the analytes. Some existing reaction rate coefficients were modified to enable the model to match laboratory and field campaign determinations of ion-cluster sensitivities as functions of CIMS sample flow rate and ambient humidity. Relative trends in predicted and observed sensitivities are compared as instrument specific factors preclude a direct calculation of instrument sensitivity as a function of sample pressure and humidity. Predicted sensitivity trends and experimental sensitivity trends suggested the model captured the reagent ion and cluster chemistry and reproduced trends in ion-cluster sensitivity with sample flow and humidity observed with a CIMS instrument developed for atmospheric peroxide measurements (PCIMS). The model was further used to investigate the potential for isobaric compounds as interferences in the measurement of the above species. For ambient $O_3$ mixing ratios more than 50 times those of $H_2O_2$, $O_3^-(H_2O)$ was predicted to be a significant isobaric interference to the measurement of $H_2O_2$ using $O_2^-(H_2O_2)$ at m/z 66. $O_3$ and $NO$ give rise to species and cluster ions, $CO_3^-(H_2O)$ and $NO_3^-(H_2O)$, respectively, which interfere in the measurement of $CH_3OOH$ using $O_2^-(CH_3OOH)$ at m/z 80. The $CO_3^-(H_2O)$ interference assumed one of its $O$ atoms was $^{18}O$ and present in the cluster in proportion to its natural abundance. The model results indicated monitoring water vapor mixing ratio, m/z 78 for $CO_3^-(H_2O)$ and m/z 98 for isotopic $CO_3^-(H_2O)_2$ can be used to determine when $CO_3^-(H_2O)$ interference is significant. Similarly, monitoring water vapor mixing ratio, m/z 62 for $NO_3^-$ and m/z 98 for $NO_3^-(H_2O)_2$ can be used to determine when $NO_3^-(H_2O)$ interference is significant.



**Keywords:** Chemical ionization mass spectrometry, multi-reagent ion, hydrogen peroxide, methyl peroxide, formic acid, acetic acid, cluster-ion chemistry, negative ion kinetics, model.

## 1 Introduction

Atmospheric measurements of hydrogen peroxide ($H_2O_2$), methyl peroxide ($CH_3OOH$), formic acid (hereafter referred to as $HFo$), and acetic acid (hereafter referred to as $HAc$) have evolved over the past half century. Current state-of-the-art measurements use chemical ionization mass spectrometry (e.g., Crounse et al., 2006; de Gouw and Warneke, 2007; Veres et al., 2008; St. Clair et al., 2010; Le Breton et al., 2012; Lee et al., 2014; Baasandorj et al., 2015; O'Sullivan et al., 2017; Treadaway et al., 2017) with a variety of reagent ions (e.g., $H_3O^+$, $O_2^-$, $CF_3O^-$, $I^-$, $CH_3C\langle O\rangle O^-$, $O_2^-\langle CO_2\rangle$). O'Sullivan et al. (2017) and Treadaway et al. (2017) have presented a hybrid reagent ion instrument for the simultaneous measurement of the peroxides and organic acids. Here an ion-neutral chemical kinetic model is described and used to simulate the negative-ion chemistry occurring within their mixed-reagent gas chemical ionization mass spectrometer (PCIMS). The "P" is derived from the instrument's original configuration to measure $H_2O_2$ and $CH_3OOH$ (O'Sullivan et al., 2017), and which was later modified to quantify $HFo$ and $HAc$ (Treadaway, 2015; Treadaway et al., 2017).

The PCIMS instrument and basic ion cluster schemes are described in O'Sullivan et al. (2017), Treadaway (2015), and Treadaway et al. (2017). Serendipity led to the use of a mixed reagent gas stream composed of nitrogen ($N_2$), oxygen ($O_2$), carbon dioxide ($CO_2$) and iodomethane ($CH_3I$). $O_2$ and $CO_2$ reagent gases provided $O_2^-$, $O_2^-(O_2)$ and $O_2^-(CO_2)$ as reagent ions. $CH_3I$ reagent gas yielded iodide ions ($I^-$). The PCIMS instrument was specifically designed as an aircraft flyable instrument and was flown on the NCAR-HIAPER aircraft in the Deep Convective Clouds and Chemistry experiment (DC3; Barth et al., 2016) and on the NCAR C-130 aircraft in the Front Range Air Pollution and Photochemistry Experiment (FRAPPE; https://www2.acom.ucar.edu/frappe). In these programs, a fixed area critical orifice was used on the sample inlet to PCIMS. Consequently, the air sample flow rate into the instrument varied with ambient sample pressure and analyte sensitivity (defined as the cluster ion counts per second per analyte reaction cell mixing ratio, e.g., cps/ppb), varied with ambient pressure. As documented for many atmospheric CIMS instruments (e.g., Slusher et al., 2004; Crounse et al., 2006; St. Clair et al., 2010; Le Breton et al., 2012; Lee et al., 2014; Baasandorj et al., 2015), analyte sensitivity was dependent upon the reaction cell water vapor mixing ratio. The humidity and pressure sensitivity dependencies were complex and explored in the laboratory to improve calibration. The objective of this paper is to present a model chemical mechanism which provides a theoretical basis to investigate the influences of ambient pressure (variable sample flow and reagent ion carrier gas flow rates), water vapor and other trace gases: ozone ($O_3$), nitric oxide ($NO$), nitrogen dioxide ($NO_2$) and nitric acid ($HNO_3$) on ion-cluster sensitivities for $H_2O_2$, $CH_3OOH$, $HFo$, and $HAc$. The model is extensible to simulating the negative ion chemistry of other reagent gas, ion source and reaction cell or drift-tube systems.



## 2 Methods

**2.1 PCIMS Instrument**

The physical PCIMS instrument is described in O'Sullivan et al. (2017), Treadaway (2015) and Treadaway et al. (2017). A physical description of the instrument and calibration schemes are presented in the Appendix. The instrument flow and electronic configuration described in the Appendix was used throughout the field and laboratory work reported here. The PCIMS m/z range was 1-500 m/z and the mass resolution was 1.0 m/z. The main component effecting ion-cluster

transmission through the system was the collision dissociation chamber (CDC) consisting of an entrance plate and octopole ion guide. The CDC plate DC voltage and the octopole DC and RF voltages were adjusted to maximize the transmission of the hydroperoxide analyte cluster ions $O_2^-(CO_2)(H_2O_2)$ and $O_2^-(CH_3OOH)$ and to reduce the signal from other ions near their respective masses. From an analysis of the hydration reactions for:

$$O_2^-(H_2O)_{n-1} + H_2O \rightleftarrows O_2^-(H_2O)_n; \quad n = 1 - 5,$$

$$CO_3^-(H_2O)_{n-1} + H_2O \rightleftarrows CO_3^-(H_2O)_n; \quad n = 1 - 3,$$

$$NO_3^-(H_2O)_{n-1} + H_2O \rightleftarrows NO_3^-(H_2O)_n; \quad n = 1 - 2,$$

and

$$I^-(H_2O)_{n-1} + H_2O \rightleftarrows I^-(H_2O)_n; \quad n = 1 - 4,$$

their PCIMS signals and their thermodynamic reaction enthalpy, entropy and Gibb's energy, we did not observe expected

signals for $O_2^-(H_2O)_{n>3}$, $CO_3^-(H_2O)_{n>2}$ and $I^-(H_2O)_{n>1}$ given the signals for the preceding hydrate and the reaction cell water vapor pressure mixing ratio. This corresponded with an absence of the hydrates whose standard reaction enthalpy is nominally more positive than -50 kJ mol[-1] and whose standard reaction Gibb's energy is nominally more positive than -20 kJ mol[-1]. Thermodynamic data were from NIST Chemistry WebBook SRD69 (Bartmess, 2016).

PCIMS uses a mixed reagent ion chemistry: $O_2^-$, $O_2^-(CO_2)$ and $I^-$ to produce cluster ions with $H_2O_2$ [$O_2^-(H_2O_2)$, $O_2^-(CO_2)(H_2O_2)$, $I^-(H_2O_2)$ at masses 66, 110 and 161, respectively], with $CH_3OOH$ [$O_2^-(CH_3OOH)$, mass 80], with HFo [$I^-(HFo)$, mass 173] and HAc [$I^-(HAc)$, mass 187]. There is a weaker signal for $I^-(CH_3OOH)$ at mass 175 but it is not considered in the model. For completeness, its sensitivity as a function of reaction cell water mixing ratio is shown in Fig. S1 and its sensitivity relative to $I^-(H_2O_2)$, $I^-(HFo)$ and $I^-(HAc)$ is shown in Fig. S2 of the Supplemental Information.

In flight, ambient air was sampled through a heated probe held at 30°C in DC3 and 70°C in FRAPPE. The higher temperature in FRAPPE was used to partially alleviate an inlet contamination issue. Sample air is passed from the inlet to the instrument using heated PFA ® Teflon tubing. All "wetted" surfaces from the probe to the physical instrument are PFA ® Teflon. The inlet system is pumped by the instrument's vacuum system and by a second scroll pump (Varian model IDP-

95 3) to increase sample airflow though the inlet tubing improving response time and ameliorating potential wall artifacts in $H_2O_2$, $CH_3OOH$, $HFo$, and $HAc$. Standard additions of hydroperoxides were performed in DC3 and hydroperoxide and



organic acid standard additions were performed in FRAPPE. The gas standards were added before a selectable entrance to two traps in series (Carulite-200 ®, Carus Corp., Peru, IL; NaOH on fiberglass wool). There was a constant flow of standard gas to within 0.3 m of the inlet and a "draw-back" line was used to divert the standard and an equal amount of sample air to

waste under normal conditions (Fig. A2b). A 2-way valve on the "draw-back" line of the syringe addition system and a 3-way valve near the instrument inlet (Fig. A2b) were used to select between 1 of 4 modes: 1) the sample air, 2) sample air with gas standard addition, 3) sample air passed through the traps as a field blank, or 4) sample air with gas standards added and passed through the traps to evaluate trap efficiency. In this way instrument calibration and trap efficiency were monitored.

$H_2O_2$, $CH_3OOH$ and $CD_3OOH$ (trideuterated methyl peroxide) gas standard additions are available for research flights 6-22 in DC3. $H_2O_2$ and $HFo$ standard additions are reported for all 15 research flights in FRAPPE. However, in FRAPPE, the instrument experienced severe vibration in flight, which caused "chatter" in the MFCs, and there was a significant contaminant in the hanger. Consequently, $CH_3OOH$ calibrations were reported for the last 11 flights after the MFC mounts

were reconfigured and "chatter" was greatly reduced. $HAc$ standard additions were only available for a portion of these FRAPPE flights as the contamination problem was minimized only on longer flights or after high altitude runs. The standard additions used here were further screened to ensure each standard addition cycle, (ambient air, ambient air with gas standards added, ambient air), was completed at constant pressure (altitude).

The laboratory calibration set up is described fully in Treadaway (2015) and only briefly here (block schematic shown in Fig. A2a). A pure-air generator (Model 737-10A, Aadco Instruments Inc., Cleves, OH) supplied the carrier air stream at 10 slpm (standard liters per minute, $T_{ref}$=0 ºC, $P_{ref}$=1013.25 hPa). This air stream was split between dry (5-10 slpm) and humidified lines (0-5 slpm) and the total flow was maintained at 10 slpm. The water concentration in the humidified line is controlled with two gas washing bottles and a gas-water equilibration coil immersed in a water bath held at either 15 or 25

120    ºC. At the latter bath temperature, it was necessary to reset the room temperature from 22 to 30 ºC to prevent condensation in the line. For some experiments, gas standard additions were performed with an external Henry's Law type equilibration coil with concurrent aqueous flow at 0.4 mL min⁻¹, air flow at 0.4 slpm, gas and aqueous flows are separated at the end of the coil using a cyclone separator, and the coil-cyclone are immersed in a water bath held at 15 ºC. The Henry's Law system was plumbed to the carrier air stream after the humidification line. A needle valve was used to simulate lower ambient

pressures (Fig. A2a) as in flight. The aircraft standard addition system was also used and this remained plumbed downstream of the laboratory air pressure control system. Air pressures between 120 and 1013 hPa were sampled (nominally set at 120, 180, 300, 600 and 1013 hPa). By changing the proportion of air flow through the dry (10, 9, 8, 7, 6 and 5 slpm) and humidified lines (0, 1, 2, 3, 4, and 5 slpm) and the inlet pressure, it was possible to alter the reaction cell water vapor mixing ratio from 30 to 20,000 ppm.



**2.2 Ion-Neutral Chemical Mechanism**

The chemical mechanism is guided by the ion-neutral reaction suites and kinetic summaries of Albritton (1978), Huertas et al. (1978) Ikezoe et al. (1987), Turunen et al. (1996), Kazil (2002), Popov (2010), Kovács et al. (2016) developed to simulate ion-neutral chemistry of the atmosphere. Necessary modifications and extensions of the chemical mechanisms to fit the PCIMS sensitivities are described here and in more detail within the Supplemental Information.

Some trace components of ambient air can compete for the reagent ions and ion-neutral clusters effecting the yield (sensitivity) of the analyte ion clusters of interest and as well as forming isobaric interfering ion or ion clusters. Analyte ion-clusters and identified potential interfering ion species at specific m/z ratios are listed in Table 1. Also listed are primary ion cluster m/z ratios used to assess potential isobaric interferences. For example, m/z 78 is used to monitor $CO_3^-(H_2O)$, which in turn is used to estimate the potential interference at m/z 80 from $CO_3^-(H_2O)$ should one of its four O atoms be a mass 18 stable isotope of oxygen, $^{18}O$, and present at its natural abundance of 0.204%.

Time-dependent concentrations of 73 species (neutrals, ions and ion-clusters), listed in Appendix Table A1, are predicted in time according to the 209 bi- and ter-molecular reactions presented in Appendix Table A2. Analyte cluster ion formation reactions are relisted in Table A3 for clarity. Potential isobaric interference ion cluster formation reaction sequences are listed in Table A4 for clarity, as well. A set of 72 ordinary differential equations was solved using the *ode23t* solver (MatLab version R2016b, The MathWorks, Inc.) with relative tolerance equal to $3x10^{-9}$ and the absolute tolerance equal to $3x10^{-12}$.

**2.3 Reaction Rate Coefficients**

Reaction rate coefficients are taken from Popov (2010), Kazil (2002), Ikezoe et al (1987), Kawamoto and Ogawa (1986), Fahey et al. (1982), Albritton (1978), Huertas et al. (1978), Fehsenfeld and Fergusson (1974), Adams et al. (1970), Fehsenfeld et al. (1969; 1967), and references therein. Reaction rate coefficient units are $s^{-1}$, $cm^3$ $molec^{-1}$ $s^{-1}$, and $cm^6$ $molec^{-2}$ $s^{-1}$ for uni-, bi- and ter-molecular reactions, respectively. Equilibrium constants determined using reaction Gibbs energy, have been converted appropriately assuming (ideal gas behavior, $T_{ref} = 298.15$ K, $P_{ref} = 1013.25$ hPa, and $N_{ref} = 2.46x10^{25}$ molec $m^{-3}$). Most of the reaction rate coefficients were experimentally determined and a few were theoretically estimated (e.g., Kazil, 2002; Iyer et al., 2016). However, some of the rate constants listed in the above compilations were simply presumed (e.g., Mohnen, 1972; Huertas et al., 1978) and these presumptions have carried forward into later works. Several rate coefficients were estimated from the Gibb's reaction energy, $\Delta G_{rxn}^o$, or equilibrium constant, $K_{eq}$, with either a measured forward or reverse reaction rate coefficient following Albritton (1978), *i.e.*, $K_{eq} = \frac{k_{for}}{k_{rev}}$ and $K_{eq} = e^{\{\Delta G_{rxn}^o/(RT_o)\}}$. The majority of Gibbs reaction energies are taken directly from the NIST Chemistry WebBook (Bartmess, 2016). Generally available neutral, ion and ion-cluster formation enthalpy, entropy and Gibb's formation energy for the $O_2^- - O_2 - CO_2 -$



$H_2O$ − hydroperoxide system are listed in Table A5 in the Appendix. Reaction enthalpy, reaction entropy and Gibb's reaction energies for this system are listed in Table A6 in the Appendix. Notes on the development of the thermodynamic Tables A5 and A6 are given in supplement information section 1.1. As called out below and in the supplemental information, care is required in applying $K_{eq} = \frac{k_{forward}}{k_{reverse}}$ as the implied reaction system may not represent a simple concerted reaction pair in equilibrium but involve a reaction sequence in steady-state. For several of the ion-hydrate cluster reactions:

$$X^-(H_2O)_{n-1} + H_2O + M \leftrightarrow X^-(H_2O)_n + M \qquad K_{eq} = \frac{k_{hydration}}{k_{dehydration}} = e^{\{\Delta G^o_{rxn}/R/T_o\}}$$

neither the forward termolecular hydration rate constant ($k_{hydration}$) nor the bimolecular dehydration rate constant ($k_{dehydration}$) is known. In this case, rate coefficients are estimated from the observation that a strong correlation exists between the log of $k_{dehydration}$ and the Gibbs energy of the hydration reaction ($\Delta G^o_{rxn}$). The correlation is shown in Fig. 1a. Known dehydration rate coefficients include those that are experimentally determined by direct measurement of $k_{dehydration}$ and those that are estimated from the hydration equilibrium constant and a measured $k_{hydration}$. For the cases in which neither $k_{hydration}$ or $k_{dehydration}$ is known, $k_{dehydration}$ is first estimated using $\Delta G^o_{rxn}$ as its predictor (i.e., the linear regression model "fit" in Fig. 1a) and $k_{hydration}$ is subsequently estimated from the predicted $k_{dehydration}$ value and $K_{eq}$. Fig. 1b shows known $k_{hydration}$ and estimated $k_{hydration}$ plotted as a function of $\Delta G^o_{rxn}$. Note further that, with only a few exceptions, individual $k_{hydration}$ rates fall within a factor of two of the mean value (dashed green line) and are near the collision limit. A factor of two falls within the accepted uncertainty estimated for the reaction rate coefficients. The uncertainties in ion-molecule reaction rate coefficients as reported by their original authors are included in the summary by Ikezoe et al. (1987). Typically, reaction rate coefficient uncertainty is reported to be a factor of two (e.g., Albritton, 1978; Fahey et al., 1982; Ikezoe et al., 1987). Although for a few reactions, "best" reaction rate coefficient uncertainties of ±20% can be found (e.g., Ikezoe et al., 1987). Here a factor of two is taken as the uncertainty in the reaction rate coefficients. Additional notes on the development of reaction rate coefficients are given in sections 1.2-1.4 of the supplemental information.

The following generic equilibrium reaction sequences, after, e.g., Crounse et al. (2006) and Le Breton et al. (2011), are used to describe negative ion, $X^-$, cluster formation with an analyte, $A$, representing $H_2O_2$, $CH_3OOH$, $HFo$, and $HAc$:

$$X^- + H_2O + M \leftrightarrow X^-(H_2O) + M \qquad\qquad (R1)$$

$$X^- + A + M \leftrightarrow X^-(A) + M \qquad\qquad (R2)$$

$$X^-(H_2O) + A \leftrightarrow X^-(A) + H_2O \qquad\qquad (R3)$$

Reactions R1-R3 correspond to our reactions (22) - (45). As discussed below, we have added poly-hydrate "switching" type reactions:

$$X^-(H_2O)_n + A \leftrightarrow X^-(A)(H_2O)_{n-1} + H_2O \qquad\qquad (R4)$$

to account for observed higher order humidity effects on $O_2^-$, $O_2^-(CO_2^-)$ and $I^-$ hydroperoxide and organic acid sensitivity. The kinetics of R1 are discussed above. For the most part, our ion-analyte cluster reaction kinetics are unstudied. Measured reaction rate coefficients were available for $H_2O_2$ clustering with $NO_2^-$, $NO_3^-$, $Cl^-$, and $HSO_4^-$ (Böhringer et al., 1984). Iyer




et al. (2016) using *ab initio* methods estimated reaction rate coefficients and binding energies for $I^-$ with $HFo$ and $HAc$.

They also calculated binding energies for $I^-$ reactions with $H_2O_2$ and $CH_3OOH$ (Iyer, Pers. Comm., 2017). The calculated binding energies for $I^-(HFo)$, $I^-(HAc)$, $I^-(H_2O_2)$, and $I^-(CH_3OOH)$ were 100, 73, 70, and 60 kJ mol[-1], respectively. Iyer et al. predicted sensitivities for the Lee et al. (2014) instrument using the calculated binding energies and measured sensitivities. We have normalized these to $I^-(HFo)$ and the predicted relative sensitivities were 1.000, 0.034, 0.007, and 0.001 for $I^-(HFo)$, $I^-(HAc)$, $I^-(H_2O_2)$ and $I^-(CH_3OOH)$, respectively. These were consistent with the observations of

O'Sullivan et al. (2017) in which they noted observing $I^-(H_2O_2)$ and sometimes $I^-(CH_3OOH)$ clusters with the PCIMS instrument and with Treadaway et al. (2017) in which they observed a weak standard addition calibration signal for $I^-(CH_3OOH)$ during FRAPPE and in the laboratory in preparation for FRAPPE (unpublished; in the supplemental information Fig. S1), and relative to those for $I^-(HFo)$, $I^-(HAc)$, and $I^-(H_2O_2)$ (supplemental information Fig. S2). In contrast to the prediction of Iyer, Treadaway et al. observed comparable sensitivities for $I^-(HAc)$ and $I^-(H_2O_2)$.

At the constant reaction cell instrument pressure of 22 hPa, the forward rate coefficient for R2 was taken to be constant and the reaction and its rate coefficient were given as a pseudo-bimolecular reaction with an initial reaction rate coefficient of $3 \times 10^{-9}$, which is near the bi-molecular collision limits calculated by Kazil (2002) and Iyer et al., (2016). This reaction rate coefficient was presumed for R3, as well, although in the literature switching reaction rate coefficients on the order of $10^{-10}$

are also used as estimates.

The reverse reaction coefficient of R2 is estimated using the assumed forward rate constant and the equilibrium constant for R2. As noted above, reaction Gibbs energies and equilibrium constants are available for R1 (Bartmess, 2016). A more limited set of Gibbs energies and equilibrium constants are available for R2 with $H_2O_2$ as the analyte (Böhringer et al., 1984;

Cappa et al., 2001; Messer et al., 2000; O'Sullivan et al., 2017). Following Böhringer, we have used known reaction enthalpies, $\Delta H^\circ_{R5}$, or reaction Gibbs energies, $\Delta G^\circ_{R5}$, of the ion protonation reaction, R5,

$$X^- + H^+ \leftrightarrow XH \qquad\qquad (R5)$$

as linear predictors of the reaction Gibbs energy for R1 and R2, $\Delta G^\circ_{R1} \ and \ \Delta G^\circ_{R2}$ and $\Delta G^\circ_{R2}$, (and therefore the equilibrium constant) with $A = H_2O_2$ and $X^- = O_2^-$, $O_2^-(CO_2)$ or $I^-$. Fig. 2a illustrates the linear relationships between $\Delta G^\circ_{R5}$ with $\Delta G^\circ_{R1}$

for $H_2O$, and with $\Delta G^\circ_{R2}$ for $H_2O_2$. Fig. 2b shows the linear relationship between $\Delta G^\circ_{R5}$ and $\Delta G^\circ_{R3}$, where $\Delta G^\circ_{R3} = \Delta G^\circ_{R2} - \Delta G^\circ_{R1}$. The predicted equilibrium constants, $K_{R2}$, for $O_2^-$, $O_2^-(CO_2)$ and $I^-$ are $3.2 \times 10^{16}$, $3.5 \times 10^7$, and $1.4 \times 10^7$ (atm[-1]), respectively. The coefficients of determination were the same regardless of whether $\Delta H^\circ_{R5}$ or $\Delta G^\circ_{R5}$ was used to predict $\Delta G^\circ_{R1}$ or $\Delta G^\circ_{R2}$ and subsequently $\Delta G^\circ_{R3}$. As noted in the supplemental information, there is some question as to whether $O_2^-(CO_2)(H_2O)$ follows a simple reaction pair or involves a more complex set of reactions at steady state and a linear

prediction of $\Delta G^\circ_{R3}$ could be an oversimplification and a source of error for reaction rate constants involving this species.





The kinetics and equilibrium constants for $CH_3OOH$ ion cluster formation are more speculative. Cappa et al. (2001) using *ab initio* methods have estimated $\Delta H^{\circ}_{R2}$ and $\Delta G^{\circ}_{R2}$ for cluster formation with $CO_3^-$, -69 and -34 kJ/mol, respectively, and $\Delta H^{\circ}_{R3}$ and $\Delta G^{\circ}_{R3}$ for R3, -17 and -9 kJ/mol, respectively. Iyer (Pers. Comm., 2016) estimated the $CH_3OOH$ binding energy

with $I^-$ is -60 kJ/mol. Messer et al. (2000) also using *ab initio* methods examined the kinetics and energetics of $H_2O_2$ and $CH_3OOH$ cluster ion formation with $F^-$. They reported theoretical collision-limit rate coefficients of $1.42 \times 10^{-9}$ and $1.47 \times 10^{-9}$, respectively, for reactions with $F^-(H_2O)_3$. Their theoretical rate coefficients were bracketed by their experimental determined rates of $0.96 - 1.92 \times 10^{-9}$. The $F^-(H_2O)_3$ ion was the predominant reagent ion under their experimental humidity conditions that gave rise to an ion-peroxide signal. Messer et al. further stated the rate of reaction was relatively unchanged

for $F^-$ hydration numbers less than 6. Payzant and Kebarle (1972), Fehsenfeld and Ferguson (1974), and Fahey et al. (1982) discussed reaction rates of $O_2^-$ with variable numbers of water molecules attached and indicated they varied only slightly with different extents of hydration. We have therefore assumed the reaction rate coefficients for hydrated $O_2^-$ ions with $H_2O_2$ and $CH_3OOH$ do not vary significantly with hydration. The forward rate constants for R2, R3 and R4 are set at near the collision rate for $H_2O_2$ and $CH_3OOH$. As a caveat, we note some switching reaction rate coefficients for less tightly bound neutral species, e.g., $O_2, CO_2, H_2O$, are reported to be on the order of $10^{-10}$.

Water vapor is commonly added to the $CH_3I$ reagent gas stream in $I^-$ based CIMS instruments because it enhances sensitivity for some analytes (e.g., Slusher et al., 2004; Le Breton et al., 2011). Whether this is because $H_2O$ is a better third body energy carrier, such as in R2, or adds a switching reaction, R3, to the instrument's development of an $I^-(A)$ cluster ion

is not clear, though discussions point to the latter. Per Iyer et al. (2016), the R2 forward reaction rate coefficient is initially set at the collision limit for $HFo$ and $HAc$. The forward reaction rate coefficients for these two compounds in R3 were set initially at the collision limit. Last, the hydration equilibrium constants for $I^-$ are such that under our laboratory and field experimental conditions, $I^-$ and $I^-(H_2O)$ dominate over $I^-(H_2O)_{n>1}$ ions. Even so at the highest humidities studied it was necessary to include R4 for n=2, but inclusion of R4 with n>2 was unnecessary even when the R4 reaction rate coefficient

was set at the collision limit.

Last, Iyer et al. (2016) examined the probability of collisional stabilization of $HFo$ (atom number 5) compared to maximum sensitivity molecules (atom number >8) and found the former gave sensitivities dependent of reaction cell pressure, whereas, the latter were independent of pressure. Our analytes have between 4 and 6 atoms and our use of collision limit reaction rate

coefficients could have resulted in an over prediction of the rates of $I^-$ cluster formation.



## 2.4 Model Assumptions

Individual model runs are performed in two stages. The first simulated the chemistry of the ion source region (alpha emitter, and reagent ion gas mixture). The product ion outflow of the source was then instantaneously mixed with the sample air stream and the ion-neutral chemistry of the reaction cell was simulated second. The following assumptions were made:

1) alpha particle emission was uniform along the ion source tube length.

2) the ions directly generated by the alpha particles passing through >99% $N_2$ gas consisted solely of $e^-$ and $N_2^+$ ions; the mechanism included several negative ions and $N_2^+$ is the only positive ion considered.

3) the energy of a $^{210}$Po alpha particle is 5.3 MeV; the formation enthalpy of a $N_2^+$ and $e^-$ ion pair from $N_2$ gas is 34 eV; thus, as a zeroth order estimate, a 20 mCi $^{210}$Po alpha source (the stated activity of the NRD P-2130 Electrostatic

Eliminator ®) generated on the order of $10^{14}$ ion pairs per second.

4) ion and neutral molecule concentrations varied along the flow direction and were radially uniform in the ion source tube and reaction cell.

5) gas fluid flow in the ion source tube and in the reaction cell followed plug-flow.

6) the reagent gas stream and ambient air stream were mixed instantaneously and uniformly at the point of contact.

7) ion clusters containing $N_2$, $O_2$, and $CO_2$ as neutrals were not considered with the exceptions of $O_2^-(CO_2)$ and $O_2^-(O_2)$.

8) wall effects on negative ions, neutral species and heterogeneous chemistry were ignored. The first assumption is supported by the fact the ion source tube and reaction cell walls have a -2V bias applied.

9) the negative ion positive ion recombination was parameterized using a single pseudo positive ion, "$N_2^{+}$", that reacts with each negative ion and whose rate constant followed Kazil (2002) and was tracked through:

$$N_2^+ + X_i^- \xrightarrow{k=6x10^{-8}(300/_T)+1.25x10^{-25}[M](300/_T)} neutral\ products \qquad (157_i)$$

where, $157_i$ indicated the reaction was included for each of the negative ion species.

## 2.5 Initial Concentrations

The initial reagent gas mixture was composed of $N_2$, $O_2$, $CO_2$, and $CH_3I$ in proportions that vary with sample air pressure and sample-air flow rate. The total flow rate through the reaction cell was constant at 4.68 slpm (standard liters per minute;

$T_{ref}$ = 0 °C, $P_{ref}$ = 1013.25 hPa). The ambient air sample flow rate varied (range 0.3 to 3 slpm) with ambient pressure (range 120 to 1013.25 hPa) as did the $N_2$ flow rate through the ion-source tube (range 2.1 to 4.3 slpm). Reagent gas concentrations in the ion-source tube varied accordingly and their initial concentrations for different sample pressures and flow conditions are listed in Table 2. Six representative pressures are shown, which span the range of pressures encountered in the DC3 and FRAPPE airborne field campaigns and in the laboratory work. The ion source tube and reaction cell temperature and

pressure were taken to be 25 °C and 22 hPa, respectively. The model chemical system was then integrated in time for the length of the gas transit time through the ion source tube.



Ambient air was then mixed with the reagent ion stream. Ambient air, for the purpose of defining reaction cell ion concentration and analyte ion-molecule cluster concentration, included $N_2$ (~79 %), $O_2$ (~21 %), $CO_2$ (~400 ppm), $CH_4$ (~2 ppm), $H_2$ (~0.5 ppm), $N_2O$ (~0.32 ppm), $O_3$ (~0.05 ppm), NO (~ 1 ppb), $NO_2$ (~1 ppb), $HNO_3$ (~1 ppb), $H_2O_2$ (~1 ppb), $CH_3OOH$ (~1 ppb), HFo (~1 ppb) and HAc (~1 ppb). The noble gases, carbon monoxide, and other oxygenated volatile organic compounds were not considered here. The air-sample water vapor mixing ratio was varied from 10 to 31700 ppm to span the range found in the troposphere. Simulation results are presented as a function of reaction cell water vapor mixing ratio and ambient sample pressure.

## 3 Model Results

The development of ions along the length of the ion-source tube for representative ambient pressures of 1013 and 307 hPa is illustrated in Fig. 3 for the fully developed model. The total ion density was at or near steady-state approximately 2/3s of the way through the ion source tube, although $O_2^-(CO_2)$ increased throughout the length of the source tube at the expense of $O_2^-$ and $O_2^-(O_2)$ (blue traces). Distance along the length of the source tube is displayed on the x-axis instead of time because the time of transit through the tube varied with air-sample pressure.

In the ion-source tube, electrons ($e^-$) were captured by $O_2$ and dissociatively captured by $CH_3I$:

$$e^- + O_2 \xrightarrow{M} O_2^- \tag{1), (2}$$

$$e^- + CH_3I \rightarrow I^- + CH_3 \tag{3}$$

The $M$ indicates a third molecule participates in the reaction. A portion of the initial $O_2^-$ reacted with $O_2$, $CO_2$ and $CH_3I$ in the source tube and in the reaction cell yielding secondary $I^-$ and $O_2^-(CO_2)$ and $O_2^-(O_2)$ cluster ions:

$$O_2^- + CH_3I \rightarrow\rightarrow I^- + O_2 + CH_3 \tag{4}$$

$$O_2^- + CO_2 \xrightarrow{M} O_2^-(CO_2) \tag{5}$$

$$O_2^- + O_2 \xrightarrow{M} O_2^-(O_2) \tag{55}$$

Note: reaction numbers follow their order within the model reactions presented in Table S4 in the supplemental information. Reaction (4) was inferred based on $O_2^-$ reactivity with $CH_3F$, $CH_3Cl$, $CH_3Br$, $CF_4$, $CF_3Cl$, $CF_3Br$, and $CF_3I$ (Fehsenfeld et al., 1975; Streit, 1982; McDonald and Chowdhury, 1985; Grimsrud, 1992; Morris, 1992; Kazil, 2002). At high $CH_3I$ mixing ratios such as those used in Slusher et al. (2004) and Le Breton (2012) without $O_2$ or $CO_2$, $CH_3I$ initially captured the electrons and $I^-$ was the primary negative reagent ion generated within the ion source tube. For our reagent mixture, the model indicated approximately 20% of the initial electrons lead to $I^-$ formation and 80% to $O_2^-$ and its clusters. The secondary formation of $I^-$ from $O_2^-$ was small. At the end of the ion source tube, the concentrations of the primary reagent ions: $O_2^-$, $O_2^-(CO_2)$ and $I^-$, were predicted at comparable concentrations.





In the termolecular reactions above, others below, and in Appendix Table A2, M represented the concentration of all other
gases, mostly $N_2$ followed by $O_2$, $H_2O$, $Ar$ and $CO_2$, whereas in the experiments used to determine reaction rate constants, M
usually represented a single predominant gas like $O_2$, $CO_2$, $Ar$ or $He$. $N_2$ or $H_2O$ were seldom included as the third body. In
the case of electron attachment, Pack and Phelps (1966) noted faster rates of $e^-$ attachment with $H_2O$ as the third body
compared to $O_2$ or $CO_2$ and that rates with these gases were faster than those when $N_2$ was the third body. Under humid
conditions in the reaction cell section, this was a potential source of error, larger than the factor of two given above.
Electron concentrations at the end of the source tube were predicted, under our assumptions, to be a factor three larger than
any of the other reagent ions.

Fig. 4 shows the predicted concentrations of $I^-$, $O_2^-$, $O_2^-(CO_2)$, $O_3^-$ and $CO_3^-$ ions, ion-hydrates and analyte-ion clusters along
the length of the reaction cell after the ion source stream was mixed with the sample air stream for the fully developed
model. In Fig. 4, distance along the reaction cell was used for the x-axis for consistency with Fig. 3, although the transit
time through this section was constant and time or distance were equivalent. The reaction cell transit time was 17.8 ms.
Two representative simulations are shown, one with an atmospheric pressure and subsequent sample flow rate
commensurate with 1013 hPa and air sample water vapor mixing ratio of 17800 ppm (16 °C dew point temperature) and the
other with a sample pressure of 307 hPa, a commensurate sample flow rate and water vapor mixing ratio of 1000 ppm (−32
335    °C frost point temperature). The corresponding reaction cell water vapor mixing ratios were ~9700 ppm and ~130 ppm,
respectively. The ions and ion-hydrates were at or near steady-state approximately 1/3 of the way down the reaction cell
length. The ion-analyte clusters increased steadily down the length of the cell (Fig. 4e), with the exception of $I^-(H_2O_2)$ and
$I^-(HAc)$ and possibly $O_2^-(H_2O_2)$, which peaked at 1 to 3 cm and then decline with distance down the remainder of the
reaction cell. The was attributed to the time needed to form the clusters of interest and their titration after formation by ion-
ion recombination with $N_2^+$. This suggested a longer flow tube could improve sensitivity for those ions which have not
reached their maximum value by the end of the reaction cell but at the expense of those clusters which have already peaked.

The ambient ozone mixing ratio was set to 50 ppb in all cases show here. No appreciable difference in $O_2^-(H_2O_2)$,
$O_2^-(CH_3OOH)$, $O_2^-(CO_2)(H_2O_2)$, $O_2^-(H_2O_2)$, $I^-(HFo)$ and $I^-(HAc)$ sensitivity was observed when the assumed sample $O_3$
mixing ratio was halved, doubled, or set to 500 ppb. The latter was apropos to the UTLS (upper troposphere lower
stratosphere). Simulated hydroperoxide and organic acid sensitivities were relatively unchanged even with a 10-fold
increase in $O_3$. However, as will be discussed later, $O_3$ influenced potential isobaric interferences at m/z 66, $O_3^-(H_2O)$, and
80, $^{18}O$ of $CO_3^-(H_2O)$ and the changes in $O_3$ resulted in a nearly proportionate increase in these ions. As an aside to $O_3^-$ and
$CO_3^-$ chemistries, O'Sullivan et al. (2017) proposed to use $CO_3^-$ as a hydroperoxide reagent ion following the work of Cappa
and Elrod (2001) but were unsuccessful in its implementation. Our simulations indicated O'Sullivan's $O_3$ reagent
concentrations were likely too low.



Fig. 5a-10a show experimentally determined sensitivities as a function of reaction cell water vapor mixing ratio for $O_2^-(H_2O_2)$, $O_2^-(CH_3OOH)$, $O_2^-(CO_2)(H_2O_2)$, $I^-(H_2O_2)$, $I^-(HFo)$ and $I^-(HAc)$, respectively.  The field and laboratory calibration data were primarily dependent upon humidity and secondarily on sample ambient pressure.  The experimental data were first binned by humidity irrespective of ambient or sample pressure. The horizontal bar of the plus symbol denotes the limits of a reaction cell water vapor mixing ratio bin and is plotted at the mean sensitivity for that bin.  The length of the vertical bar of the plus symbol indicates one standard deviation of the bin and the variability was due to variations arising from pressure, ambient concentrations during the standard addition, systematic variations due to water vapor across a bin, calibration gas precision and instrument precision. The yellow shaded portions outlined the experimentally determined sensitivity from laboratory experiments and field calibrations from DC3 and FRAPPE.

Fig. 5b-10b show the model simulated ion-analyte cluster sensitivities at the end of the reaction cell as a function of reaction cell water vapor mixing ratio and ambient sample pressure for the fully developed model.  The simulated ion-analyte cluster sensitivities are expressed in arbitrary units as Lee et al. (2014) and Iyer et al. (2016) have argued instrumental factors make it nearly impossible to map simulated instrument sensitivity to that determined experimentally.  However, assuming instrumental process effects were proportional for each individual ion-neutral cluster, instrument sensitivity trends with pressure and water vapor for each ion-neutral should be captured by the simulations and scalable on an individual basis.  "Sensitivity" as shown is the ion-cluster concentration divided by the analyte's ambient mixing ratio, 1 ppb, and was expected to be proportional to counts per ppb.  These were further scaled to a maximum of 1 by dividing the predicted sensitivities by the maximum sensitivity calculated for that cluster regardless of sample pressure or humidity.

## 4 Discussion

The initial model mechanism including $O_2^-$, $O_2^-(H_2O)$, $O_2^-(CO_2)$, $I^-$ and $I^-(H_2O)$ alone was unable to simulate the sample pressure and water dependent trends in sensitivity for $O_2^-(H_2O_2)$, $O_2^-(CH_3OOH)$, $O_2^-(CO_2)(H_2O_2)$, $I^-(H_2O_2)$, $I^-(HFo)$ and $I^-(HAc)$, not shown. The simulated $O_2^-$ hydroperoxide cluster species showed too peaked of a dependence on water vapor. The increase in sensitivity with increasing water vapor was too steep at reaction cell water vapor mixing ratios less than 100 ppm and the decrease in sensitivity with increasing water vapor at reaction cell water vapor mixing ratios greater than 1000 ppm was also too steep. Higher-order ion hydrates, $(H_2O)_{n>1}$, carbonates, $(CO_2)_{n>1}$, mixed hydrate-carbonates, $(H_2O)_m(CO_2)_n$, and their chemical reactions were added to rectify this (see supplemental information for details).





The mechanism so modified remained insufficient to reproduce the $O_2^-$ and $O_2^-(CO_2)$ peroxide sensitivities as a function of reaction cell $H_2O$ mixing ratio and ambient sample pressure (not shown) observed in the field and laboratory measurements. Multiple avenues were explored including (see Supplemental Information):

1. modification of reaction rate coefficients for reactions (12), (13), (14), (21), (24), (147), and (148), which describe the $O_2^- - (H_2O) - (CO_2)$ switching system,

2. the inclusion of reactions (149) - (152) allowing for higher order hydrates to form the analyte cluster ions for peroxides,

3. the inclusion of $O_3$ reactions with $O_2^-$ to yield $O_3^-$ and subsequent $CO_3^-$ ions,

4. invoking a new carbonation reaction (199),

$$O_2^-(H_2O_2) + (CO_2) + M \rightarrow O_2^-(CO_2)(H_2O_2) + H_2O + M \qquad (199)$$

5. the addition of reactions (204) - (209) allowing for higher order hydrates of $I^-$ to form the analyte cluster ions for hydrogen peroxide and the organic acids.

The addition of $O_3$ chemistry had little effect on sensitivity, whereas the first two changes improved the pressure dependent sensitivity and water vapor trends for the $O_2^-$ hydropreoxide clusters but did not significantly improve the pressure and water vapor sensitivity trends in $O_2^-(CO_2)(H_2O_2)$, necessitating a process like reaction (199).

Under low water vapor conditions the model under predicted $O_2^-(H_2O_2)$ and $O_2^-(CH_3OOH)$ at higher sample pressures relative to low sample pressures. This was primarily due to the conversion of most of the $O_2^-$ reagent ion to $O_2^-(CO_2)$ in the absence of water vapor (e.g., Kebarle et al., 1972). The conversion of $O_2^-$ to $O_3^-$ was of minor influence. Significantly, at higher water vapor mixing ratios, the $O_2^-$ hydroperoxide clusters and the $O_2^-(CO_2)(H_2O_2)$ cluster were under predicted because of $O_2^-$ hydrate formation, $O_2^-(H_2O)_{n=1,5}$ and the switching reactions included in the initial mechanism:

$$O_2^-(H_2O) + H_2O_2 \rightarrow O_2^-(H_2O_2) + H_2O \qquad (28)$$

$$O_2^-(H_2O) + CH_3OOH \rightarrow O_2^-(CH_3OOH) + H_2O \qquad (36)$$

with rate coefficients set at approximately the collision limit (3x10$^{-9}$) were unable to simulate enough hydroperoxide cluster formation.

A solution to the low humidity problem was suggested by the work of Fehsenfeld and Ferguson (1974), Fahey et al. (1982) and Böhringer et al. (1984). Böhringer et al. suggested there is a hierarchical shift in cluster ion ligands, $X^- \cdot L$, according to ion-ligand bond energy ($E_{bond}$), for a specific ion. Their ordering followed: $E_{bond}(X^- \cdot H_2O) < E_{bond}(X^- \cdot SO_2) < E_{bond}(X^- \cdot H_2O_2) < E_{bond}(X^- \cdot HCl) < E_{bond}(X^- \cdot HNO_3)$. Adams et al., (1970), Fehsenfeld and Ferguson (1974) and Fahey et al. (1982) presented and discussed the thermodynamics and kinetics of $O_2^-$, $O_2^-(O_2)$, $O_2^-(CO_2)$, $O_2^-(H_2O)_{n=1,3}$ and $O_2^-(CO_2)(H_2O)_{n=1,3}$. Their bond energies suggested the series given by Böhringer et al. could be extended to include O$_2$ and CO$_2$ as ligands with O$_2$ more weakly bound than H$_2$O and with CO$_2$ and H$_2$O being comparable bound, such that:





$$E_{bond}(X^- N_2) < E_{bond}(X^- O_2) < E_{bond}(X^- H_2O) \approx E_{bond}(X^- CO_2) \quad < \quad E_{bond}(X^- SO_2) \quad < \quad E_{bond}(X^- H_2O_2) \quad \approx$$

$E_{bond}(X^- CH_3OOH) < E_{bond}(X^- HCl) < E_{bond}(X^- \cdot HNO_3)$. Last, *ab initio* calculations suggested the ligand bond energy

of $CH_3OOH$ lies above $H_2O$ and near or just below $H_2O_2$ (Cappa and Elrod, 2001) or well above that of both $H_2O$ and $H_2O_2$

(O'Sullivan et al., 2017). Consequently, we have speculated that both peroxides may readily switch with the $CO_2$ in

$O_2^-(CO_2)$ and have included the following two $CO_2$ – peroxide "switching" reactions in the model mechanism:

$$O_2^-(CO_2) + H_2O_2 \rightarrow O_2^-(H_2O_2) + CO_2 \tag{160}$$

$$O_2^-(CO_2) + CH_3OOH \rightarrow O_2^-(CH_3OOH) + CO_2 \tag{161}$$

From sensitivity studies varying forward rate constants, $k_{160}$ and $k_{161}$, rate coefficient values of $2 \times 10^{-12}$ and $1 \times 10^{-12}$ cm$^{-3}$

molec$^{-1}$ s$^{-1}$, respectively, or greater were sufficient to remove the pressure dependent discrepancy in $O_2^-$ peroxide sensitivity

noted at low water vapor mixing ratios. The magnitude of these rate coefficients was reasonable given the bonding energy

progression and the rate coefficients reported by Adams et al. (1970) for $CO_2$ – $H_2O$ switching reactions:

$$O_2^-(H_2O) + CO_2 \rightarrow O_2^-(CO_2) + H_2O \qquad\qquad k_{12}=\sim 6 \times 10^{-10} \tag{12}$$

$$O_2^-(CO_2) + H_2O \rightarrow O_2^-(H_2O) + CO_2 \qquad\qquad k_{13}=\sim 2 \times 10^{-10} \tag{13}$$

where, $k_{13}=k_{12}/K_{eq}(2.3)$, and those estimated by Fahey et al. (1982) for:

$$O_2^-(H_2O)_2 + CO_2 \rightarrow O_2^-(CO_2)(H_2O) + H_2O \qquad\qquad k_{147}=7 \times 10^{-11} \tag{147}$$

and

430 $$O_2^-(CO_2)(H_2O) + H_2O \rightarrow O_2^-(H_2O)_2 + CO_2 \qquad\qquad k_{148} \sim 1 \times 10^{-9} \tag{148}$$

Finally, a new hybrid clustering reaction was invoked:

$$O_2^-(H_2O_2) + (CO_2) + M \rightarrow O_2^-(CO_2)(H_2O_2) + M \qquad\qquad k_{199}=5.0 \times 10^{-29} \tag{199}$$

which finally enabled the modified chemical mechanism to resolve the pressure and water vapor trends in $O_2^-$ and $O_2^-(CO_2)$

peroxide sensitivities as shown in Figs 5b-7b. This reaction has a calculated Gibb's reaction energy of 7 kJ mol$^{-1}$ (see

supplemental information) and therefore is not spontaneous. However, as discussed within the supplement, the reaction is

expected to be exothermic, many of the formation and reaction enthalpy and Gibb's energy terms used to derive this value

were from experimental data having factor of two uncertainties in rates and 5 to 10 kJ mol$^{-1}$ uncertainties. A Gibb's reaction

energy of 7 kJ mol$^{-1}$ is at the 6[th] significant figure of the *ab initio* calculations and may be near the limit of the theoretical

calculations of O'Sullivan et al. (2017) from which it was derived.

The pressure and humidity trends exhibited in the laboratory $CH_3I$ experiments and in the field calibrations for $H_2O_2$ (Fig. 8;

DC3 and FRAPPE); $HFo$ (Fig. 9; FRAPPE) and $HAc$ (Fig. 10; FRAPPE) could not be reconciled using only monohydrate

switching reactions (30-33, 38-41, and 42-45). The decrease in sensitivity with increasing humidity was too steep (not

shown), and hypothesized reactions (204-209) were added. These reactions likely summarize multistep sequences such as:

$$I^-(H_2O)_n + A \rightarrow I^- A(H_2O)_{n-1} + H_2O$$





…

$$I^- A(H_2O) \xrightarrow{M} I^- A + H_2O$$

where $A$ represents $H_2O_2$, $HFo$ or $HAc$. It should be noted that reactions (205), (207) and (209) involving $I^-(H_2O)_3$ were inconsequential in reconciling the observed pressure and humidity trends in sensitivity even when reaction rate coefficients were set equal to those of reactions (204), (206), and (208). Consequently, the reaction rate coefficients for the $I^-(H_2O)_3 + A$ reactions were unconstrained by our analyses.

The families of $O_2^-$ and $I^-$ concentrations from the full-model chemical mechanism are shown in Fig. 11 as a function of reaction cell water vapor mixing ratio for several sample pressures. Sample flow rate (sample pressure) had virtually no effect on $I^-$ and $I^-(H_2O)_n$ concentrations, whereas, the reaction cell water vapor mixing ratio has a very strong effect on $I^-(H_2O)_n$ at all levels and on $I^-$ when the mixing ratio is above $10^3$ ppm. Sample flow rate and reaction cell water vapor mixing ratio have a profound effect on $O_2^-(H_2O)_n$ and $O_2^-(CO_2)_n$ speciation and their concentrations. It was the steep drop in $O_2^-(CO_2)$ at water vapor mixing ratios greater than $10^2$ ppm, which necessitated the invocation of a reaction like (199) to predict the experimental sensitivity trends observed at m/z 110 for $O_2^-(CO_2)(H_2O_2)$.

The $I^-(H_2O_2)$ sensitivity was a critical test point in adjusting the laboratory and FRAPPE $CH_3I$ concentrations to best match the sensitivity observed in DC3 (Treadaway, 2015; Treadaway et al., 2017). The measurement objective in DC3 was the quantification of $H_2O_2$ and $CH_3OOH$ and it was conducted without organic acid field standards. During DC3 it was recognized there were quantifiable but uncalibrated signals at spectral locations attributed to $I^-(HFo)$ and $I^-(HAc)$ (Treadaway, 2015; Treadaway et al. 2017). Post mission calibrations and calibrations during the FRAPPE field campaign with hydroperoxides provided an estimate of the $CH_3I$ reagent concentration as evidence by $I^-(H_2O_2)$ and its trends with pressure and humidity (Fig. 8). The modeled trends in $I^-(H_2O_2)$ reinforce this and provide collaborative data supporting the extrapolation of the laboratory and FRAPPE $HFo$ and $HAc$ sensitivities to DC3 (Treadaway, 2015; Treadaway et al., 2017).

Many of the reaction rate coefficients used to model analyte cluster ion sensitivity trends with water vapor and pressure have been estimated by ourselves and others. This introduces uncertainties to the results but within the constraints of the calibration data. The reaction rate coefficient for reaction (38) is used as an example of the constraints placed on the rate of (38) in the context of reactions (38) - (41), the I⁻ hydrate reactions (17), (18), (78) - (81) and the inferred higher hydrate switching reactions (206) and (207). The estimated rate was $1.5 \times 10^{-10}$ cm³ molec⁻¹ s⁻¹. Doubling the rate constant for (38) led to a flattening of the linear trend in $I^-(HFo)$ sensitivity with water vapor at low water vapor mixing ratios and too sharp a peak at the maximum sensitivity. Halving the reaction rate coefficient for (38) excessively steepened the trend in $I^-(HFo)$ sensitivity with water vapor. The reaction rate coefficient for reaction (40) would have to exceed the bimolecular rate limit in order to increase the reaction rate coefficient for (39) and maintain the observed sensitivity trend. The same would also





apply to reaction (206) needed to broaden the maximum in sensitivity in $I^-(HFo)$ at 10,000 ppm water vapor. The reaction

rate coefficient for (38) could be reduced but would require proportionate reductions in the reaction rate coefficients for the

linked reactions to maintain the modeled sensitivity trend. A halving or doubling was used for the purpose of illustration as

this was within the range of uncertainty quoted for many of the measured reaction rate coefficients in the literature.

The model was further used to examine the potential for isobaric interference at m/z 80, where $CH_3OOH$ is observed, and at

m/z 66, a potential m/z for $H_2O_2$ (Table 1). Fig. 12a shows the predicted concentrations of $O_2^-(H_2O_2)$, $^{18}O$ of $NO_2^-(H_2O)$,

$O_3^-(H_2O)$, and $^{18}O$ of $O_2^-(O_2)$ all of which would appear at m/z 66. The interference ion production pathways are outlined in

the Appendix, Table A4. Fig. 12b shows the predicted concentrations of $O_2^-(CH_3OOH)$, $^{18}O$ of $CO_3^-(H_2O)$, and $NO_3^-(H_2O)$,

which would all be observed at m/z 80. The simulations assumed the ambient air contained $NO$, $NO_2$, $HNO_3$, $H_2O_2$,

$CH_3OOH$, $HFo$ and $HAc$ at mixing ratios of 1 ppb for each. $O_3$ and $CO_2$ were assumed to be 50 ppb and 400 ppm,

respectively. The ambient water vapor mixing ratio and sample pressure were varied from 10 to $3 \times 10^5$ ppm and from 120 to

1013 hPa, respectively. Note that while reaction cell water vapor pressures from 1 to $10^5$ ppm can be prescribed, the higher

mixing ratios at the lower sample pressures simulated were unrealistic owing to the decrease in air temperature, maximum

dew point temperature and consequent decrease in maximum air saturation vapor pressure with decreasing air pressure

(increasing altitude). The dotted lines (left panel) indicate the contribution to m/z 66 from $O_3^-(H_2O)$ which was proportional

to the $O_3$ mixing ratio but which passed through a maximum with respect to water vapor, increasing at lower humidities due

to (19) and then decreasing at higher humidities due to (72). At the highest sample pressure, $O_3$=200 ppb yields a maximum

predicated interference of ~$1.6 \times 10^6$ A.U. at $2 \times 10^3$ ppm of water and was comparable to the predicted signal from 1 ppb

$H_2O_2$. Increasing $NO_2$ to 100 ppb lead to a $^{18}O$ of $NO_2^-(H_2O)$ predicted interference comparable to the signal for 1 ppb

$H_2O_2$. Note reductions in $H_2O_2$ mixing ratios increase the potential for interference by these other gases. $^{18}O$ of $O_2^-(O_2)$ is

not predicted to be a significant interference at any of the pressures or humidities examined. The above discussion assumes

PCIMS does not inherently discriminate between clusters due to instrumental factors.

The measurement of $CH_3OOH$ at m/z 80 is predicted to suffer from interference by both $^{18}O$ of $CO_3^-(H_2O)$, and $NO_3^-(H_2O)$.

$O_3$ through reaction sequence (8), (9), and (15) leads to the formation of $CO_3^-(H_2O)$ and a predicted inference was

proportional to the $O_3$. The predicted interference as a function of $H_2O$ mixing ratio passed through a maximum, increasing

at first due to (15) and then decreasing due to (53). $NO$ primarily through reactions with $O_3^-$ or $O_2^-(H_2O)_{n \geq 1}$ produced

$NO_3^-$, which went on to form a hydrate. The predicted $NO_3^-(H_2O)$ interference also passed through a maximum as a

function of water vapor due to reactions (115) and (116). At reaction-cell water-vapor mixing ratios greater than a few

hundred ppm, $O_3 = 50$ ppb and $NO = 1$ ppb gave predicted interference signals comparable to the signal from 1 ppb of

$CH_3OOH$. In the DC3 project after research flight 12 and throughout FRAPPE, m/z 78 was monitored for $CO_3^-(H_2O)$ to

ensure an $^{18}O$ isotope of this compound did not appreciably interfere in the measurement of $CH_3OOH$. Similarly, m/z 62, 78





and 98, corresponding to $NO_3^-$, $CO_3^-(H_2O)$ and $NO_3^-(H_2O)_2$ or $^{18}O$ of $CO_3^-(H_2O)_2$ were monitored to ensure $NO_3^-(H_2O)$ did

not appreciably interfere in the $CH_3OOH$ measurement. As above this assumes PCIMS does not inherently discriminate

between clusters due to instrumental factors nor does it discriminate between the oxygen isotopic clusters of $CO_3^-(H_2O)$.

Currently, experimental kinetic data to examine the isobaric interferences for $I^-(HFo)$ and $I^-(HAc)$ by ethanol and

propanol, methyl formate, or glycolaldehyde, respectively are unavailable. Treadaway (2015) and Treadaway et al. (2017)

tested ethanol and 1-propanol and 2-propanol at very low and high water vapor conditions and found the sensitivity for

$I^-(HFo)$ was 100 times that for $I^-(ethanol)$ and the sensitivity for $I^-(HAc)$ was 100 times that for $I^-$ clustering with

either 1- or 2-propanol. These relative sensitivities agreed with those predicted by Iyer (Pers. Comm., 2017) for $HFo$, $HAc$,

ethanol and 2-propanol based upon their calculated binding energies with $I^-$.

## 5 Conclusions

An ion-neutral chemical kinetic model is described and used to simulate the negative-ion chemistry occurring within a

mixed-reagent ion chemical ionization mass spectrometer (CIMS). The model established a theoretical basis for

investigation of ambient pressure (variable sample flow and reagent ion carrier gas flow rates), water vapor, ozone and

oxides of nitrogen effects on ion-cluster sensitivities for hydrogen peroxide ($H_2O_2$), methyl peroxide ($CH_3OOH$), formic

acid ($HCOOH$) and acetic acid ($CH_3COOH$). The model was builts with established mechanisms, thermodynamic data and

reaction rate coefficients and these were augmented with additional reactions with estimated reaction rate coefficients. Some

existing reaction rate coefficients were modified to enable the model to match laboratory and field campaign determinations

of ion-cluster sensitivities as functions of CIMS sample flow rate and ambient humidity. Relative trends in sensitivity were

compared as instrument specific factors preclude a direct calculation of instrument sensitivity. Predicted sensitivity trends

and experimental sensitivity trends suggest the model captured the PCIMS reagent ion and cluster chemistry and reproduced

observed trends in ion-cluster sensitivity with sample flow and humidity. The model was further used to investigate the

potential for isobaric compounds as interferences in the measurement of the above species. For ambient $O_3$ mixing ratios

more than 50 times those of $H_2O_2$, $O_3^-(H_2O)$ is predicted to be a significant isobaric interference to the measurement of

$H_2O_2$ using $O_2^-(H_2O_2)$ at m/z 66. $O_3$ and $NO$ give rise to species and cluster ions, $CO_3^-(H_2O)$ and $NO_3^-(H_2O)$, respectively,

which interfere in the measurement of $CH_3OOH$ using $O_2^-(CH_3OOH)$ at m/z 80. The $CO_3^-(H_2O)$ interference requires one

of the $O$ atoms to be the stable isotope $^{18}O$. The model results indicate monitoring water vapor mixing ratio, m/z 78 for

$CO_3^-(H_2O)$ and m/z 98 for isotopic $CO_3^-(H_2O)_2$ can be used to determine when $^{18}O$ of $CO_3^-(H_2O)$ interference is significant.

Similarly, monitoring water vapor mixing ratio, m/z 62 for $NO_3^-$ and m/z 98 for $NO_3^-(H_2O)_2$ can be used to determine when

$NO_3^-(H_2O)$ interference is significant.





**Acknowledgements:** This research was supported through grants from the United States National Science Foundation: ATM-09222886 and ATM-1063467 (DWO) and ATM-1063463 (BGH) and a contract from the Colorado Department of Public Health and the Environment (BGH). We are indebted to S. Iyer (Dept. of Chemistry, University of Helsinki, Helsinki, Finland) for graciously sharing additional ab initio calculations cited in this manuscript. The authors thank D. Tanner, G. Huey, and R. Stickle (THS Instruments, LLC) for advice and patience. We thank the members of the DC3/SEACRS and FRAPPE/DISCOVER-AQ science teams and thank the NCAR-EOL Research Aviation Facility fight crew and staff for the quality of their data and for making the flight programs successful. Disclaimer: The manuscript has not been reviewed by the Colorado Department of Public Health and Environment. The scientific results and conclusions, as well as any views or opinions expressed herein, are those of the author(s) and do not necessarily reflect the views of the Department or the State of Colorado.

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





**Table 1:** PCIMS m/z for species clusters of interest and potential isobaric interfering ions or ion clusters at the mass resolution of the quadrupole mass selector.

| m/z | Ion or ion-neutral cluster[#] |
|---|---|
| 50 | $O_2^-(H_2O)$, $^{18}O$ of $O_3^-$ |
| 60 | $CO_3^-$ |
| 62 | $NO_3^-$, $^{18}O$ of $CO_3^-$ |
| 66 | $\mathbf{O_2^-(H_2O_2)}$, $^{18}O$ of $NO_2^-(H_2O)$, $O_3^-(H_2O)$, $^{18}O$ of $O_2^-(O_2)$ |
| 76 | $O_2^-(CO_2)$ |
| 78 | $CO_3^-(H_2O)$, $^{18}O$ of $O_2^-(CO_2)$ |
| 80 | $\mathbf{O_2^-(CH_3OOH)}$, $NO_3^-(H_2O)$, $^{18}O$ of $CO_3^-(H_2O)$, $NO_2^-(H_2O_2)$, $^{18}O$ of $O_2^-(HFo)$, $O_2^-(CH_2(OH)_2)$ |
| 83 | $\mathbf{O_2^-(CD_3OOH)}$ |
| 110 | $\mathbf{O_2^-(CO_2)(H_2O_2)}$ |
| 147 | $I^-(H_2{}^{18}O)$ |
| 161 | $\mathbf{I^-(H_2O_2)}$, $I^-(^{18}O^{16}O)$ |
| 173 | $\mathbf{I^-(HFo)}$, $I^-(EtOH)$, $I^-(C^{18}O^{16}O)$ |
| 175 | $\mathbf{I^-(CH_3OOH)}$, $I^-(CH_2(OH)_2)$, $I^-(O_3)$ |
| 187 | $\mathbf{I^-(HAc)}$, $I^-(MeFo)$, $I^-(PrOH)$, $I^-(2\text{-}PrOH)$, $I^-(GA)$, |

[#]**boldface** indicates a primary PCIMS analyte ion cluster mass; "$^{18}O$ of " indicates one of the ion cluster's oxygen atoms is a
mass 18 isotope of oxygen; in $CD_3OOH$ the D represents deuterium atoms; EtOH refers to ethanol, PrOH refers to 1-
propanol, 2-PrOH refers to 2-propanol; $CH_2(OH)_2$ refers to methane-diol; MeFo refers to methyl formate; GA refers to
hydroxy acetaldehyde (aka. glycolaldehyde).



**Table 2:** Initial reagent gas flow rates and reagent gas mixing ratios at six sample air pressures.

| Sample Pressure, hPa[1] | 120 | 180 | 306 | 600 | 800 | 1013 |
|---|---|---|---|---|---|---|
| $N_2$ flow rate, slpm[2] | 4.27 | 4.23 | 3.98 | 2.99 | 2.58 | 2.05 |
| $CO_2$ in air flow rate, slpm | 0.08 | 0.08 | 0.08 | 0.08 | 0.08 | 0.08 |
| $CH_3I$ in $N_2$ flow rate, slpm | 0.0005 | 0.0005 | 0.0005 | 0.0005 | 0.0005 | 0.0005 |
| $CH_3I$, ppb[3] | 0.575 | 0.580 | 0.616 | 0.814 | 0.939 | 1.174 |
| $CO_2$ ppm[4] | 7.36 | 7.42 | 7.88 | 10.42 | 12.02 | 15.02 |
| $O_2$ ppm | 3678 | 3712 | 3941 | 5212 | 6015 | 7512 |
| $N_2$, ppm | 996322 | 996288 | 996059 | 994788 | 993866 | 992488 |

[1]pressure in hPa (hecto-Pascal, equivalent to milli-bars).

[2]slpm, standard liters per minute ($T_{ref} = 0\ °C$; $P_{ref} = 1013.25$ hPa).

[3]ppb, parts per billion (molecular mixing ratio times $10^9$).

[4]ppm, parts per million (molecular mixing ratio times $10^6$).



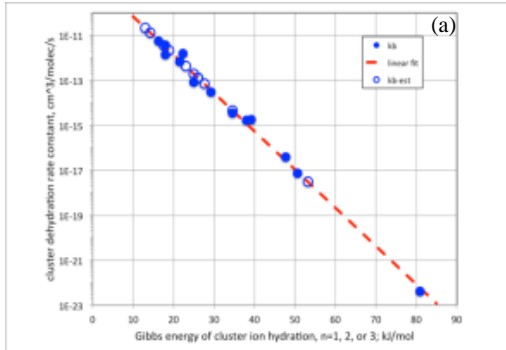
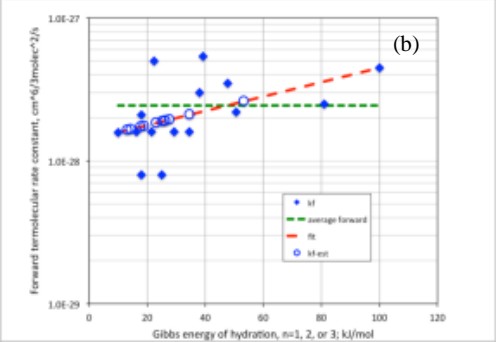

**Figure 1: Ion-cluster dehydration (a) and hydration (b) reaction rate coefficients plotted as a function of the Gibbs reaction energy ($\triangle G^o_{rxn}$) for $I^-$, $O_2^-$, $O_3^-$, $CO_3^-$, $HO^-$, $NO^-$, $NO_2^-$, and $NO_3^-$ ions for n=1-3 water molecules. Solid circles are reported rate coefficients from the literature (see text) and open circles are estimates based upon linear regression of $\ln(k_{dehydration})$ versus $\triangle G^o_{rxn}$ or $k_{hydration} = k_{dehydration} \times K_{eq}$.**





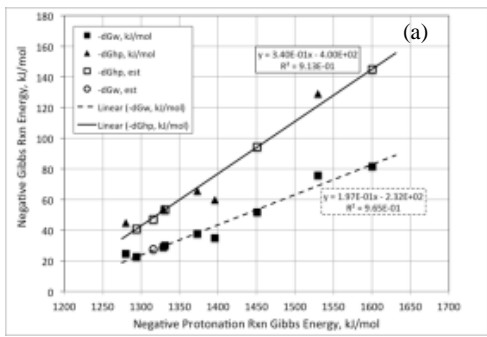
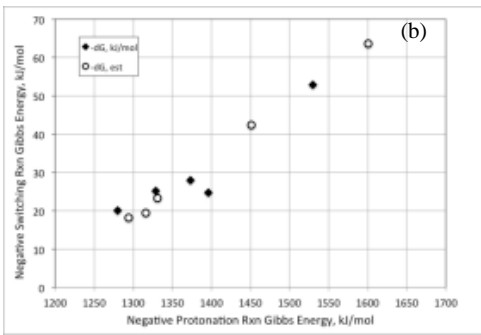

**Figure 2:** **Relationship between the ion-protonation reaction Gibbs energy, $-\Delta G^{\circ}_{R4}$, and (a) ion-hydration**

**reaction Gibbs energy, $-\Delta G^{\circ}_{R1}$, and ion-$H_2O_2$ cluster reaction Gibbs energy, $-\Delta G^{\circ}_{R2}$, or (b) $H_2O$ and $H_2O_2$**

**switching reaction Gibbs energy, $-\Delta G^{\circ}_{R3}$. Filled symbols denote measured values and open symbols indicate**

**estimated values. The lines indicate least square linear regression fits with the regression constants and**

**coefficient of determination given in the respective boxes. The use of negative values of the Gibbs energy of**

**reaction follows NIST (Bartmess, 2016) nomenclature.**




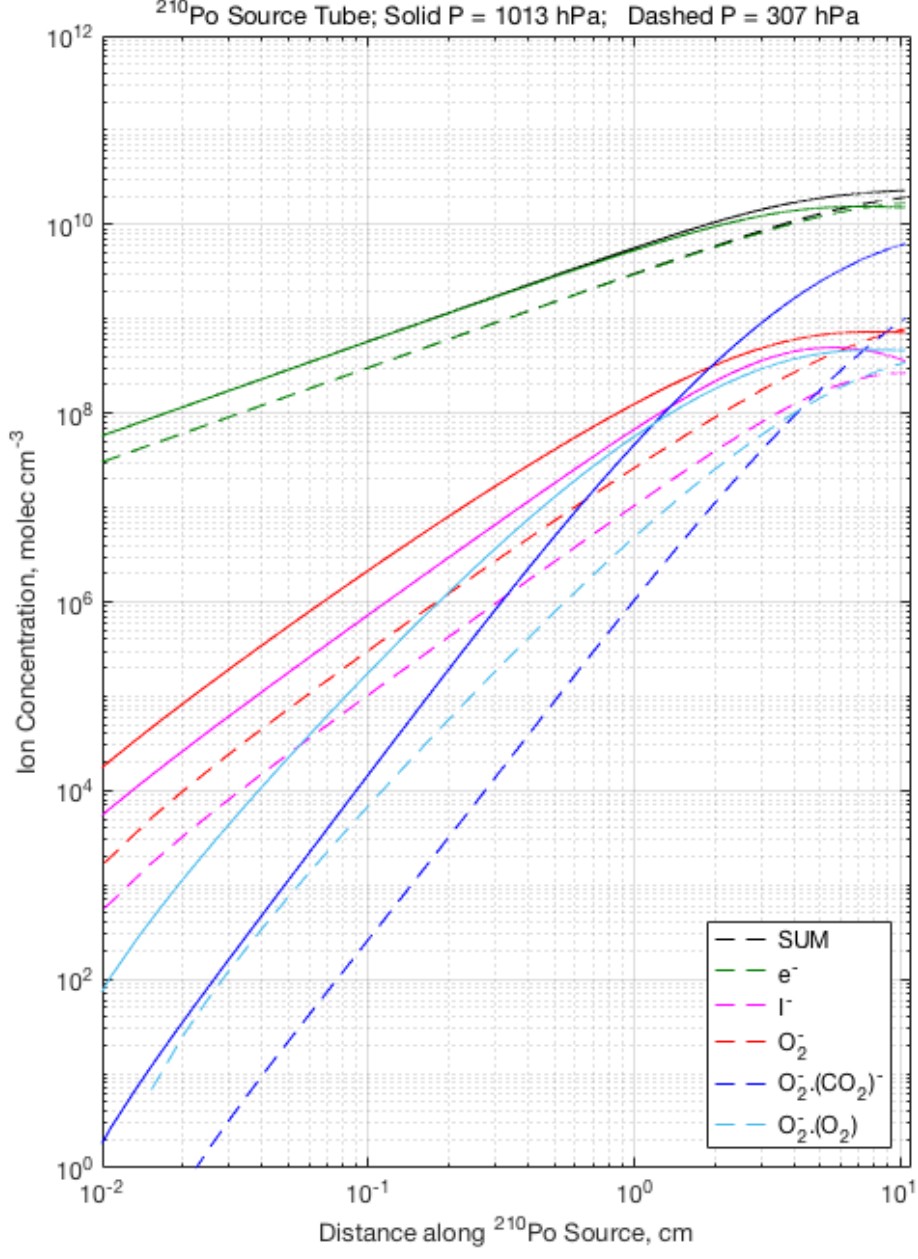

**Figure 3: Ion densities – arbitrary units – along the $^{210}$Po source tube for flow conditions of 307 hPa ambient pressure (dashed lines) and 1013 hPa (solid lines). The predominant ion after $e^-$ is $O_2^-(CO_2)$ and $I^-$ is the smallest.**



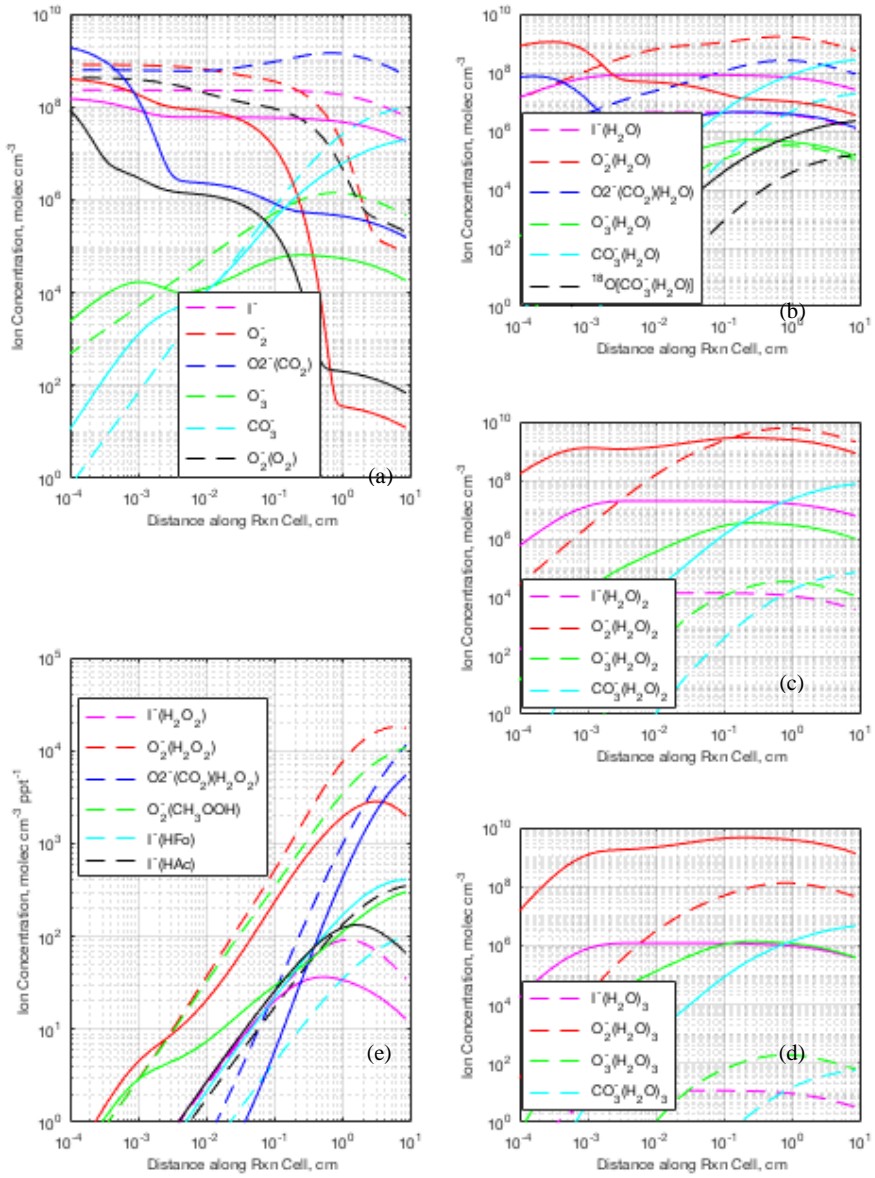

**Figure 4: Simulated ion-cluster densities along the reaction cell path for ambient pressure of $P_a$=307 hPa and reaction cell $H_2O$=133 ppm (dashed line) or $P_a$=1013 hPa and reaction cell $H_2O$=9706 ppm (solid line). Ambient $O_3$ was set equal to 50 ppb for both pressures. Reaction cell time of transit is $t_x$=17.8 ms. (a) Unhydrated reagent ion density. (b) First-hydrate reagent ion density. (c) Second-hydrate reagent ion densities. (d) Third-hydrate reagent ion densities. (e) Ion-analyte cluster ion density– arbitrary units.**



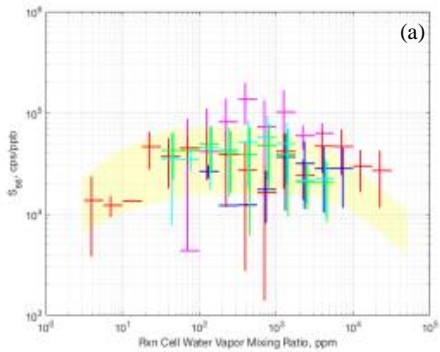
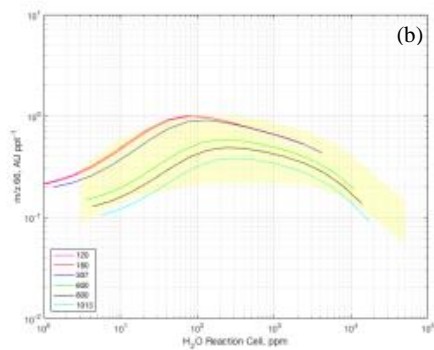

**Figure 5: (a) Experimental sensitivity (counts per second per ppb, cps/ppb) trend in $O_2^- \cdot (H_2O_2)$ as a function of reaction cell water vapor mixing ratio ($\chi_{H_2O}$, ppm), from DC3 (red), FRAPPE (blue), and the laboratory**

**[$F_{CH_3I}$ = 0.0005 (cyan) and 0.001 (green) slpm]. All calibrations were binned by $\chi_{H_2O}$ and bin widths are shown by the horizontal lines. Vertical bars indicate one standard deviation of a bin and includes sample pressure variation effects, water variation within a bin, precision of the standard addition calibration gas concentration, instrument precision and ambient mixing ratio variation across the standard addition period. There were fewer than 4 observations per bin in FRAPPE for water vapor mixing rations less than $10^3$ ppm.**

**Magenta crosses in (a) correspond to DC3 post mission calibrations without the addition of $CH_3I$ and after there had been multiple refills of the reagent $CO_2$ in air bottle. (b) normalized simulated sensitivity as a function of $\chi_{H_2O}$ for 6 different sample pressures as shown in the legend. The yellow shading maps the trends in experimental sensitivity (a) to the calculated trends in sensitivity (b) using the same normalization process.**





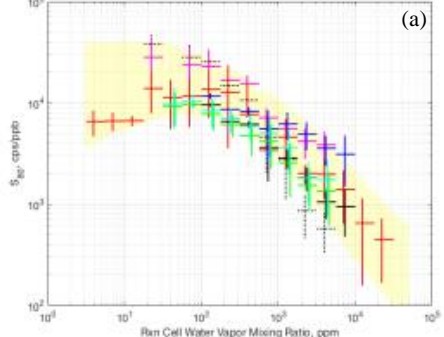
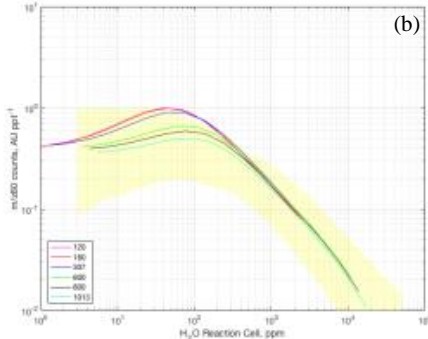

**Figure 6: Same as Fig. 5 except for O$_2^-$ (CH$_3$OOH).**





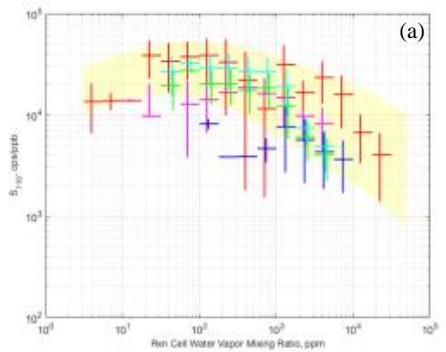

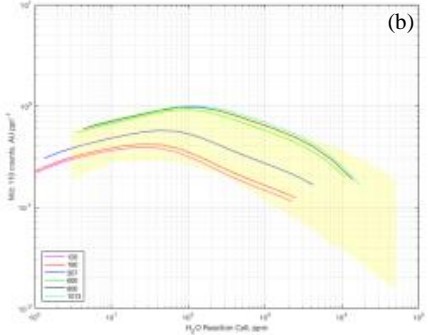

**Figure 7: Same as Fig. 5 except for $O_2^-(CO_2)(H_2O_2)$.**




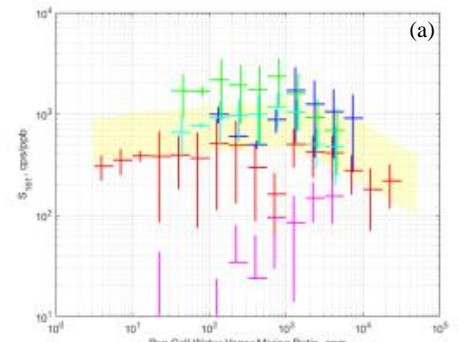

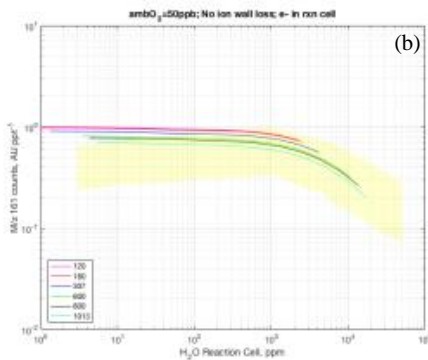

**Figure 8: Same as Fig. 5 except for $I^-(H_2O_2)$. Magenta crosses in the left panel correspond to DC3 post mission calibrations without the addition of $CH_3I$; there had been multiple refillings of the reagent $CO_2$ in air bottle and the $CH_3I$ reagent gas concentration is presumed to be very small.**





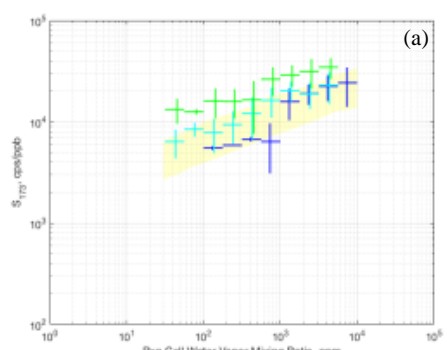
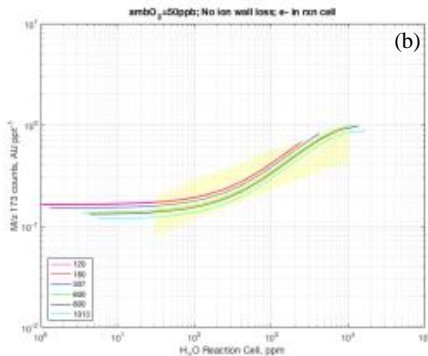

**Figure 9: Same as Fig. 5 except for I⁻ (HFo). Field calibrations for HFo were not performed in DC3.**





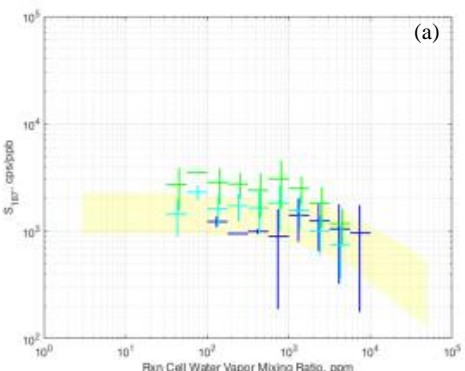
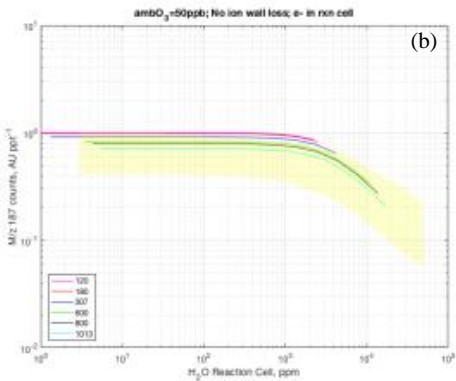

**Figure 10: Same as Fig. 9 except for I⁻(HAc). Field calibrations for HAc were not performed in DC3.**





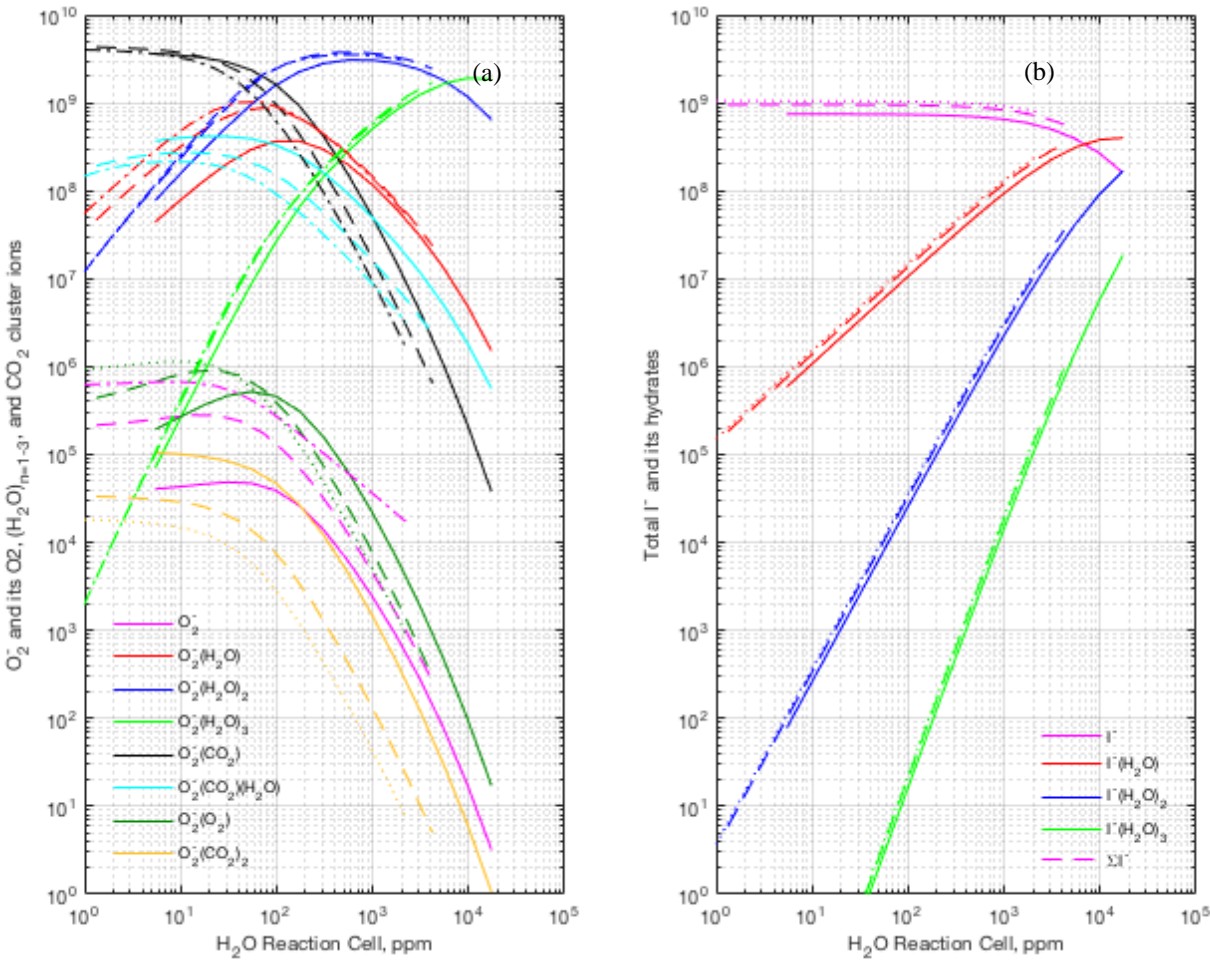

**Figure 11: (a) Oxygen anion speciation: $O_2^-$, $O_2^-(H_2O)_{n=1,3}$, $O_2^-(CO_2)_{n=1,2}$, $O_2^-(CO_2)(H_2O)$ and $O_2^-(O_2)$, as a function**

5    **of water vapor at sample pressures of 120 (dotted), 300 (dashed) and 1013 (solid) hPa. (b) As in left panel except for**

**iodide speciation: $I^-$, $I^-(H_2O)$, $I^-(H_2O)_2$ and $I^-(H_2O)_3$.**





**Figure 12: (a) Predicted ion cluster concentrations for species having m/z equal to 66: $O_2^-(H_2O_2)$ solid lines, $^{18}O$ of $NO_2^-(H_2O)$ dashed lines, $O_3^-(H_2O)$dotted lines, and $^{18}O$ of $O_2^-(O_2)$ dash-dot lines, for 6 sample pressures (hPa) indicated by color as given in the legend. (b) Predicted ion cluster concentrations for species having m/z equal to 80: $O_2^-(CH_3OOH)$ solid lines, $^{18}O$ of $CO_3^-(H_2O)$ dashed lines, and $NO_3^-(H_2O)$ dash-dot lines, for 6 sample pressures (hPa) indicated by color as given in the legend.  See text for sample air flow rate and composition and reagent gas composition used in the simulation.**



# Appendix

## A1. Instrument Details

The PCIMS instrument is shown schematically in Fig. A1 from the sample inlet to the exhaust. It was manufactured by THS Instruments, LLC (Atlanta, Georgia). The instrument consists of five separate chambers: a sample entrance chamber (labeled

5    *Split*), a reaction drift tube (*RXN cell*), a collision dissociation chamber (*CDC*), main chamber (*Main*), and the quadrupole and detector chamber (*Quad*). The entrance section allows for excess sample flow through the transfer plumbing from a sample's origin to minimize wall surface artifacts. In an aircraft, the sample probe and transfer line were heated to 35 and 70 ºC during DC3 and FRAPPE, respectively. The higher temperature in FRAPPE was used because of a ground contamination problem. The sample stream is split with the fraction entering the PCIMS reaction cell (RXN Cell) determined by a 0.51

10    mm critical orifice; the high-pressure side was set by the sample inlet pressure and the low-pressure side fixed at 22.4 hPa. The excess flow rate is regulated by an MKS Instruments (Andover, MA) 0-30 slpm mass flow controller (not shown). The reagent gas stream dynamically blends three gas streams: ultra-high purity $N_2$ (Scott-Marrin, Riverside, CA), 400 ppm $CO_2$ in ultra-high purity air (Scott-Marrin, Riverside, CA) and 5 ppm $CH_3I$ in ultrahigh purity $N_2$. These flows are regulated using MKS mass flow controllers (not shown). Representative reagent gas flow rates and mixing ratios for different sample air

pressures are listed in Table 2. The total flow through the RXN cell is 4.68 slpm and the sample flow rate can be determined by subtraction of the reagent gas flow rates from the total flow; for example, at a sample pressure of 600 hPa, the sample flow rate was 1.61 slpm. Ions are generated by passing the reagent gas stream through a commercially available Nuclecel Ionizer (Model P2031-1000, NRD LLC., Grand Island, NY), containing an $\alpha$-emitter, $^{210}Po$, with an initial activity of 20 milli-curie. The reagent ion stream is mixed at a right angle to the sample air stream, approximately 12 mm downstream

from the sample entrance orifice and 82 mm before the RXN cell to CDC chamber (collision dissociation chamber) and a pumped second port for dumping the bulk of the RXN cell reagent-sample gas stream. The RXN-to-CDC critical orifice diameter is 0.81 mm and with a high-pressure side at 22.4 hPa and a low-pressure side at 0.61 hPa, has a nominal flow rate of 0.11 slpm (http://www.tlv.com/global/TI/calculator/air-flow-rate-through-orifice.html) assuming a discharge coefficient of 1.0. The voltage on the CDC plate is -2.0 V. The PCIMS has two THS Instruments, LLC. octopole ion lenses; one set is in

the CDC (DC bias voltage = 20 V, RF voltage = 2 V) and the other is in the main chamber after the CDC (DC bias voltage = 2.49 V, RF voltage = 0.04 V). The CDC plate and first octopole voltages regulated collision energy between molecules and the cluster ions and were used to fragment weakly bound clusters. These voltages were manual adjusted to improve the signal-to-noise ratio at the m/zs of interest. The second octopole acts to focus the ions onto the entrance of the quadrupole mass selector (Extrel 19 mm rod quadrupole, controlled by an Extrel QC-150 oscillator at 2.1 MHz and a THS Instruments

control board), and the ions at a selected m/z are counted by a channeltron detector (rear plate at 3.43 kV and front plate at 1.51 kV). The pressures in the CDC and at the main chamber octopole and quadrupole chamber were 0.61, 0.0065, and 0.00011 hPa, respectively. There are two more critical orifices which separate the CDC and the main chambers, 2.08 mm,



and the main and quadrupole chambers, 2.57 mm. The nominal flow rates through these orifices are 19 and 1.3 sccm (standard cubic centimeters per second), respectively. From a mechanical perspective, 0.03% of the total flow through the RXN cell enters the quad chamber. The above instrument settings were used throughout the DC3, laboratory, FRAPPE and post FRAPPE laboratory and field work.

The m/z range of the quadrupole filter is 1-500 m/z. The PCIMS controller software provides two modes of mass selection: "hop" and "scan." In scan mode, lower and upper m/z limits, a m/z step size and the dwell time at a m/z step are defined. The smallest step size is 0.3 m/z. A typical dwell time for a scan is 50 ms. In hop mode, a variable number of fixed m/z values can be selected at an increment of 0.05 m/z. The dwell time at each fixed m/z can also be specified. In practice with

the Channeltron detector rear plate at 3.43 kV and front plate at 1.51 k, the software-hardware range limits are nominally 0 to $2 \times 10^6$ cps (counts per second) or for a 50 ms dwell time, 0 to $10^5$ counts. Random "dark" count noise in 50 ms was $\ll 1$ count or $\ll 20$ cps. The practical mass resolution of the quadrupole and detector was 1.0 m/z. This was defined as the average of the full-width at half-height of the calibration peaks for $O_2^-(H_2O_2)$, $O_2^-(CH_3OOH)$, $O_2^-(HFo)$, $O_2^-(HAc)$, $O_2^-(CO_2)(H_2O_2)$, $I^-(H_2O)$, $I^-(H_2O_2)$, $I^-(HFo)$, and $I^-(HAc)$ at 66, 78, 80, 92, 110, 145, 161, 173, and 187 m/z, respectively, and

determined using a scan step size of 0.3 m/z (mode = 0.9; range 0.9-1.2; n=9) during a HP, MHP, HFo and HAc calibration. Treadaway et al. (2017) showed the full scan from 40 to 190 m/z.

Three schemes were employed to develop calibration mixing ratios. In the laboratory and in flight, a syringe based systems inject $<10^{-9}$ m$^3$ per min. of an aqueous solution containing the species of choice in to a N$_2$ carrier stream. This flow is

constantly on. Just prior to the CIMS inlet the flow is normally diverted to waste by a "drawback" flow. This flow is turned off when calibration gas is added in to the sample stream - "standard addition". In the laboratory, two Henry's Law equilibration coils with concurrent flows of an aqueous solution containing the species of choice and an ultrahigh purity air stream as carrier. The coils are immersed in a temperature controlled water bath. These systems are show as block diagrams in Fig. A2.





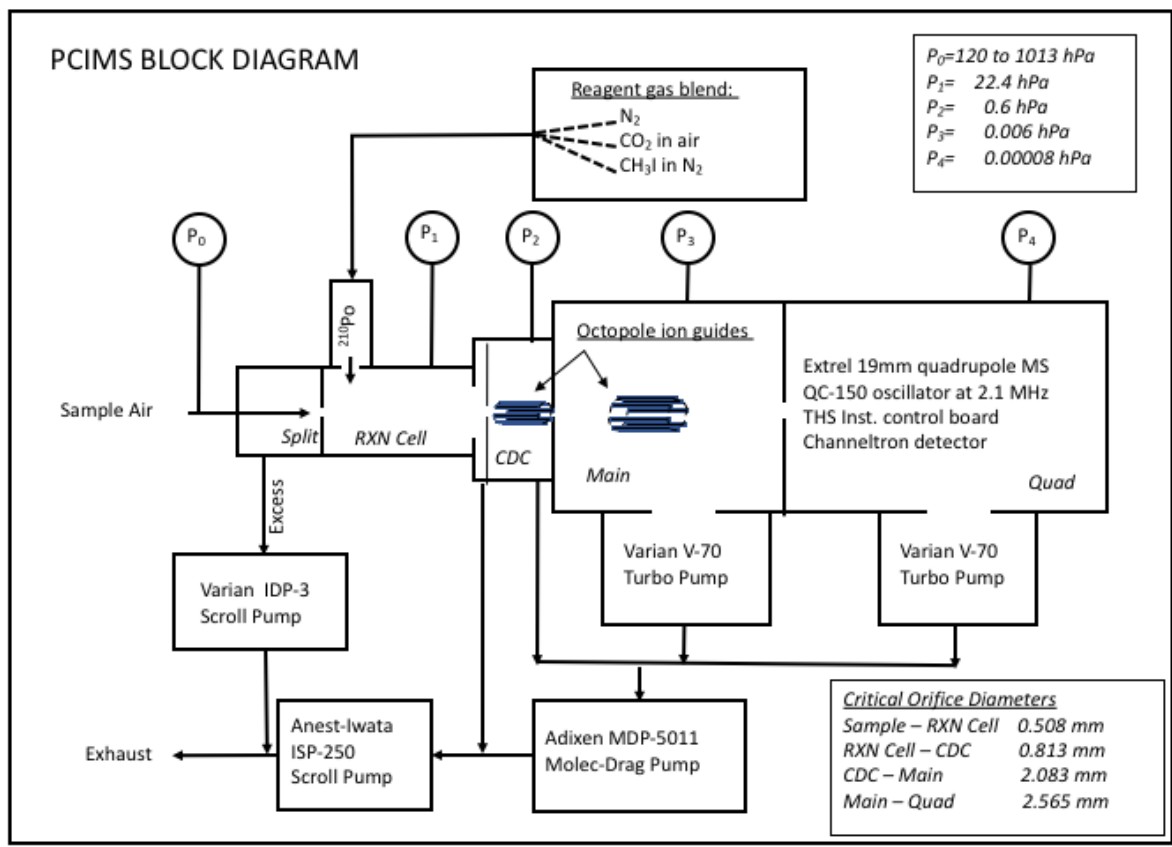

**Figure A1**: **Block schematic of the PCIMS instrument as used on the aircraft and in the laboratory, mass flow controllers are not shown. After Slusher et al. (2004), Le Breton et al. (2012), O'Sullivan et al. (2017) and Treadaway et al. (2017).**



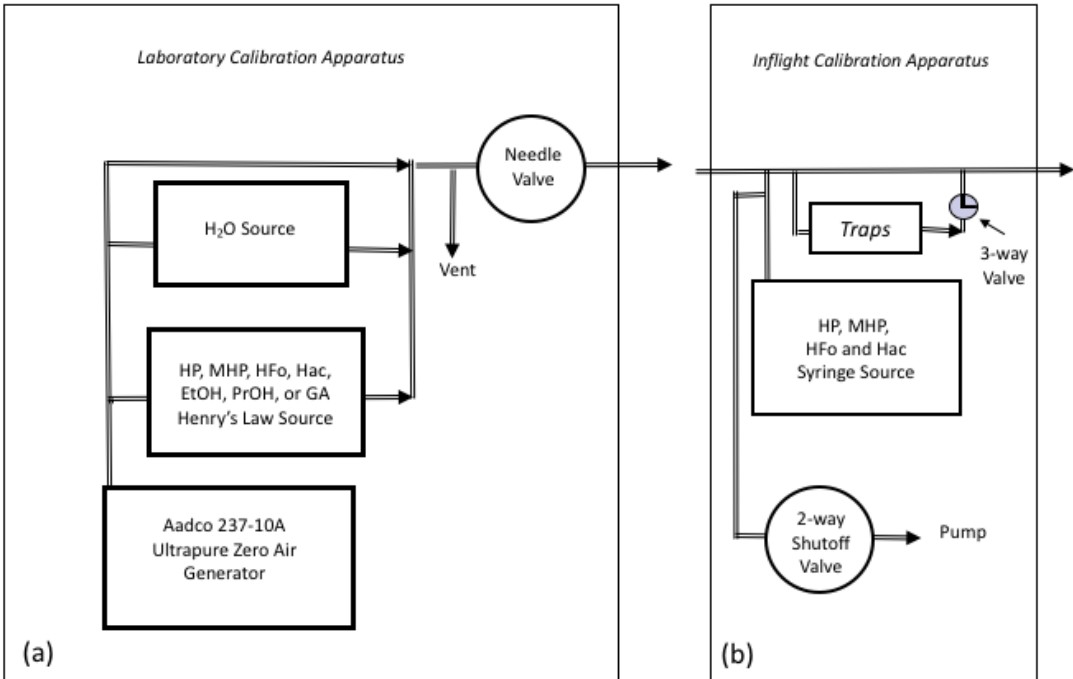

**Figure A2**: **Block schematic of laboratory (a) and in-flight (b) calibration systems, mass flow controllers are not shown. After Treadaway et al. (2017).**





**Table A1.** Ion and neutral species alphabetically sorted by species/cluster.

| Index | Species | Index | Species | Index | Species |
|---|---|---|---|---|---|
| *Neutral species* | | | | | |
| 4 | $CH_3I$ | 20 | $CH_3OOH$ | 3 | $CO_2$ |
| 21 | HFo (formic acid) | 22 | HAc (acetic acid) | 55 | $HNO_3$ |
| 16 | $H_2O$ | 19 | $H_2O_2$ | 1 | $N_2$ |
| 41 | NO | 42 | $NO_2$ | 2 | $O_2$ |
| 10 | $O_3$ | | | | |
| | | | | | |
| *Positive species* | | | | | |
| 5 | $N_2^+$ | | | | |
| | | | | | |
| *Negative Species* | | | | | |
| 12 | $CO_3^-$ | 15 | $CO_3^-(H_2O$ | 31 | $CO_3^-(H_2O)_2$ |
| 38 | $CO_3^-(H_2O)_3$ | 65 | $CO_3^-(H_2O_2)$ | 67 | $CO_3^-(CH3OOH)$ |
| 6 | $e^-$ | 34 | $HO^-$ | 59 | $HO^-(H_2O)$ |
| 60 | $HO^-(H_2O)_2$ | 61 | $HO^-(H_2O)_3$ | 7 | $I^-$ |
| 17 | $I^-(H_2O)$ | 39 | $I^-(H_2O)_2$ | 40 | $I^-(H_2O)_3$ |
| 25 | $I^-(H_2O_2)$ | 27 | $I^-(HFo)$ | 28 | $I^-(HAc)$ |
| 64 | $I^-(O_3)$ | 43 | $NO^-$ | 44 | $NO^-(H_2O)$ |
| 45 | $NO^-(H_2O)_2$ | 46 | $NO^-(H_2O)_3$ | 47 | $NO_2^-$ |
| 48 | $NO_2^-(H_2O)$ | 49 | $NO_2^-(H_2O)_2$ | 50 | $NO_2^-(H_2O)_3$ |
| 56 | $NO_2^-(H_2O_2)$ | 51 | $NO_3^-$ | 62 | $NO_3^{-*}$ or $ONOO^-$ |
| 58 | $NO_3^-(HFo)$ | 52 | $NO_3^-(H_2O)$ | 53 | $NO_3^-(H_2O)_2$ |
| 54 | $NO_3^-(H_2O)_3$ | 72 | $NO_3^-(H_2O)_4$ | 57 | $NO_3^-(H_2O_2)$ |
| 32 | $O^-$ | 33 | $O^-(H_2O)$ | 8 | $O_2^-$ |
| 26 | $O_2^-(CH_3OOH)$ | 9 | $O_2^-(CO_2)$ | 14 | $O_2^-(CO_2)(H_2O)$ |
| 69 | $O_2^-(CO_2)(H_2O)_2$ | 73 | $O_2^-(CO_2)(H_2O)_3$ | 23 | $O_2^-(CO_2)(H_2O_2)$ |
| 13 | $O_2^-(H_2O)$ | 30 | $O_2^-(H_2O)_2$ | 35 | $O_2^-(H_2O)_3$ |
| 70 | $O_2^-(H_2O)_4$ | 71 | $O_2^-(H_2O)_5$ | 24 | $O_2^-(H_2O)(H_2O_2)$ |
| 63 | $O_2^-(H_2O_2)$ | 29 | $O_2^-(O_2)$ | 11 | $O_3^-$ |
| 68 | $O_3^-(CH_3OOH)$ | 18 | $O_3^-(H_2O)$ | 36 | $O_3^-(H_2O)_2$ |
| 37 | $O_3^-(H_2O)_3$ | 66 | $O_3^-(H_2O_2)$ | | |





**Table A2:** Ion−neutral reactions sorted alphabetically by reagent ion first and then by neutral species (M denotes a third body reactant, $N_2$ or $O_2$; An index sorted list is in the supplemental information.

| Reaction | Reaction rate coefficient[1] (reference)[2] | Index |
|---|---|---|
| $CO_3^- + CH_3OOH + M \rightarrow CO_3^-(CH_3OOH) + M$ | 1.0e-28 | 170 |
| $CO_3^- + H_2O + M \rightarrow CO_3^-(H_2O) + M$ | 1.0e-28 (*FF, NIST*) | 15 |
| $CO_3^- + HNO_3 \rightarrow NO_3^- + CO_2 + HO$ | 3.5e-10(Kaz) | 144 |
| $CO_3^- + H_2O_2 + M \rightarrow CO_3^-(H_2O_2) + M$ | 1.0e-28(est) | 162 |
| $CO_3^- + NO \rightarrow NO_2^- + CO_2$ | 1.1e-11 | 83 |
| $CO_3^- + NO_2 \rightarrow NO_3^- + CO_2$ | 2.0e-10 | 84 |
| $CO_3^- + N_2O \rightarrow O_2^-(CO_2) + N_2$ | 5.e-13(*Kov*) | 69 |
| $CO_3^-(H_2O) + M \rightarrow CO_3^- + H_2O + M$ | 3.5e-14(*FF*) 3.9e-14(*Kaz*) | 16 |
| $CO_3^-(H_2O) + CH_3OOH \rightarrow CO_3^-(CH_3OOH) + H_2O$ | 1.0e-9 | 171 |
| $CO_3^-(H_2O) + H_2O + M \rightarrow CO_3^-(H_2O)_2 + M$ | 2.1e-28 | 53 |
| $CO_3^-(H_2O) + H_2O_2 \rightarrow CO_3^-(H_2O_2) + H_2O$ | 1.0e-9 | 163 |
| $CO_3^-(H_2O) + NO \rightarrow NO_2^- + CO_2 + H_2O$ | 3.5e-12 | 85 |
| $CO_3^-(H_2O) + NO \rightarrow NO_2^-(H_2O) + CO_2$ | 3.5e-12 | 86 |
| $CO_3^-(H_2O) + NO_2 \rightarrow NO_3^- + CO_2 + H_2O$ | 4.0e-11(*Kaz*) | 87 |
| $CO_3^-(H_2O) + NO_2 \rightarrow NO_3^-(H_2O) + CO_2$ | 4.0e-11(*Kaz*) | 88 |
| $CO_3^-(H_2O)_2 + M \rightarrow CO_3^-(H_2O) + H_2O + M$ | 3.7e-12(*NIST*) | 54 |
| $CO_3^-(H_2O)_2 + CH_3OOH \rightarrow CO_3^-(CH_3OOH) + 2\ H_2O$ | 1.0e-10 | 172 |
| $CO_3^-(H_2O)_2 + H_2O + M \rightarrow CO_3^-(H_2O)_3 + M$ | 1.7e-28 | 76 |
| $CO_3^-(H_2O)_2 + H_2O_2 \rightarrow\rightarrow CO_3^-(H_2O_2) + 2\ H_2O$ | 1.0e-10 | 164 |
| $CO_3^-(H_2O)_3 + M \rightarrow CO_3^-(H_2O)_2 + H_2O + M$ | 1.3e-11(*NIST*) | 77 |
| $CO_3^-(H_2O)_3 + CH_3OOH \rightarrow CO_3^-(CH_3OOH) + 3\ H_2O$ | 1.0e-11 | 173 |
| $CO_3^-(H_2O)_3 + H_2O_2 \rightarrow\rightarrow CO_3^-(H_2O_2) + 3\ H_2O$ | 1.0e-11 | 165 |
| | | |
| $e^- + CH_3I \rightarrow I^- + CH_3$ | 1.0e-7(*est*) | 3 |
| $e^- + O_2 + N_2 \rightarrow O_2^- + N_2$ | 1.0e-31(*H*) | 2 |
| $e^- + O_2 + O_2 \rightarrow O_2^- + O_2$ | 1.9e-30(*H*) | 1 |
| $e^- + O_2 + CO_2 \rightarrow O_2^- + + CO_2$ | 7/5x1.9e-30(*PP1966*) | not used |
| $e^- + O_2 + H_2O \rightarrow O_2^- + + H_2O$ | 7x1.9e-30 (*PP1966*) | not used |
| $e^- + O_3 \rightarrow O^- + O_2$ | 9e-12(*Kaz*) | 58 |
| $e^- + N_2^+ \rightarrow N_2$ | 3.6e-8(*Kaz*) | 7 |
| | | |
| $HO^- + CH_3I \rightarrow I^- + HO + CH_3$ | 3.0e-9(Ike) | 135 |
| $HO^- + CO_2 + M \rightarrow HCO_3^- + M$ | 7.6e-28 | 66 |
| $HO^- + H_2O + M \rightarrow HO^-(H_2O) + M$ | 2.5e-28 | 136 |
| $HO^- + NO_2 \rightarrow NO_2^- + HO$ | 1.1e-9 | 90 |
| $HO^- + O_3 \rightarrow O_3^- + HO$ | 9.e-10 | 67 |
| $HO^-(H_2O) + M \rightarrow HO^- + H_2O + M$ | 2.5e-28/4.2e-6(NIST) | 137 |
| $HO^-(H_2O) + H_2O + M \rightarrow HO^-(H_2O)_2 + M$ | 3.5e-28 | 138 |
| $HO^-(H_2O)_2 + M \rightarrow HO^-(H_2O) + H_2O + M$ | 3.5e-28/6.7e-12(NIST) | 139 |
| $HO^-(H_2O)_2 + H_2O + M \rightarrow HO^-(H_2O)_3 + M$ | 3.0e-28 | 140 |
| $HO^-(H_2O)_3 + M \rightarrow HO^-(H_2O)_2 + H_2O + M$ | 1.85e-13(NIST) | 141 |
| | | |
| $I^- + HAc \xrightarrow{M} I^-(HAc)$ | 7e-10(Iyer) | 42 |
| $I^- + HFo \xrightarrow{M} I^-(HFo)$ | 1.5e-10(est) | 38 |
| $I^- + H_2O + M \rightarrow I^-(H_2O) + M$ | 1.86e-28 (*Iyer, NIST*) | 17 |

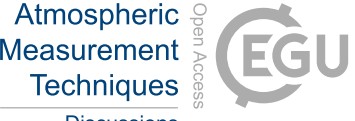



| Reaction | Reaction rate coefficient[1] (reference)[2] | Index |
|---|---|---|
| $I^- + H_2O_2 \xrightarrow{M} I^-(H_2O_2)$ | 1e-9(Iyer) | 30 |
| $I^- + HNO_3 \rightarrow NO_3^- + HI$ | 5.0e-11 | 91 |
| $I^- + O_3 + M \rightarrow I^-(O_3) + M$ | 1.0e-29(Wil) | 158 |
| $I^-(HAc) + M \rightarrow I^- + HAc + M$ | 2e-9(Iyer)/4.9e7(*NIST*) | 43 |
| $I^-(HAc) + H_2O \rightarrow I^-(H_2O) + HAc$ | 2e-9(*est*)/1.6e4(*NIST*) | 45 |
| $I^-(HFo) + M \rightarrow I^- + HFo + M$ | 1.5e-10(est)/2.01e9(*NIST*) | 39 |
| $I^-(HFo) + H_2O \rightarrow I^-(H_2O) + HFo$ | 2e-9(est)/2.21e5(*NIST*) | 41 |
| $I^-(H_2O) + M \rightarrow I^- + H_2O + M$ | 1.86e-28/2.9e-16 (*Iyer, NIST*) | 18 |
| $I^-(H_2O) + HAc \rightarrow I^-(HAc) + H_2O$ | 2e-9(*est*) | 44 |
| $I^-(H_2O) + HFo \rightarrow I^-(HFo) + H_2O$ | 2e-9(est) | 40 |
| $I^-(H_2O) + H_2O + M \rightarrow I^-(H_2O)_2 + M$ | 1.74e-28 | 78 |
| $I^-(H_2O) + H_2O_2 \rightarrow I^-(H_2O_2) + H_2O$ | 2e-9(est) | 32 |
| $I^-(H_2O)_2 + M \rightarrow I^-(H_2O) + H_2O + M$ | 3.57e-12(*NIST*) | 79 |
| $I^-(H_2O)_2 + HAc \rightarrow I^-(HAc) + 2H_2O$ | 3e-9(est) | 208 |
| $I^-(H_2O)_2 + HFo \rightarrow I^-(HFo) + 2H_2O$ | 2e-9(est) | 206 |
| $I^-(H_2O)_2 + H_2O + M \rightarrow I^-(H_2O)_3 + M$ | 2.14e-11*1.28e-17(*NIST*) | 80 |
| $I^-(H_2O)_2 + H_2O_2 \rightarrow I^-(H_2O_2) + 2H_2O$ | 2e-9(est) | 204 |
| $I^-(H_2O)_3 + M \rightarrow I^-(H_2O)_2 + H_2O + M$ | 2.14e-11 | 81 |
| $I^-(H_2O)_3 + HAc \rightarrow I^-(HAc) + 3H_2O$ | <3e-9(not used) | 209 |
| $I^-(H_2O)_3 + HFo \rightarrow I^-(HFo) + 3H_2O$ | <3e-9(not used) | 207 |
| $I^-(H_2O)_3 + H_2O_2 \rightarrow I^-(H_2O_2) + 3H_2O$ | <3e-9(not used) | 205 |
| $I^-(H_2O_2) + M \rightarrow I^- + H_2O_2 + M$ | 1e-9/6.33e5(est) | 31 |
| $I^-(H_2O_2) + H_2O \rightarrow I^-(H_2O) + H_2O_2$ | 2e-9/4.26e3(est) | 33 |
| $I^-(O_3) + M \rightarrow I^- + O_3 + M$ | 1.0e-13(Wil) | 159 |
| | | |
| $NO^- + CO_2 \rightarrow e^- + NO + CO_2$ | 8.3e-12 | 117 |
| $NO^- + H_2O + M \rightarrow NO^-(H_2O) + M$ | 2.63e-28 | 121 |
| $NO^- + NO_2 \rightarrow NO_2^- + NO$ | 7.4e-10 | 118 |
| $NO^- + O_2 \rightarrow O_2^- + NO$ | 5.0e-10 | 119 |
| $NO^-(H_2O) + M \rightarrow NO^- + H_2O + M$ | 3.05e-18(NIST) | 122 |
| $NO^-(H_2O) + H_2O + M \rightarrow NO^-(H_2O)_2 + M$ | 2.12e-28 | 123 |
| $NO^-(H_2O)_2 + M \rightarrow NO^-(H_2O) + H_2O + M$ | 4.62e-15(NIST) | 124 |
| $NO^-(H_2O)_2 + H_2O + M \rightarrow NO^-(H_2O)_3 + M$ | 1.90e-28 | 125 |
| $NO^-(H_2O)_3 + M \rightarrow NO^-(H_2O)_2 + H_2O + M$ | 1.95e-13(NIST) | 126 |
| | | |
| $NO_2^- + H_2O + M \rightarrow NO_2^-(H_2O) + M$ | 1.6e-28 | 112 |
| $NO_2^- + H_2O_2 + M \rightarrow NO_2^-(H_2O_2) + M$ | 1.0e-28(est) | 129 |
| $NO_2^- + NO_2 \rightarrow NO_3^- + NO$ | 2.0e-13 | 92 |
| $NO_2^- + N_2O \rightarrow NO_3^- + N_2$ | 1.0e-12 | 120 |
| $NO_2^- + O_3 \rightarrow NO_3^- + O_2$ | 1.2e-10 | 98 |
| $NO_2^-(H_2O) + M \rightarrow NO_2^- + H_2O + M$ | 3.53e-15(*NIST*) | 93 |
| $NO_2^-(H_2O) + H_2O + M \rightarrow NO_2^-(H_2O)_2 + M$ | 8.0e-29 | 113 |
| $NO_2^-(H_2O)_2 + M \rightarrow NO_2^-(H_2O) + H_2O + M$ | 8.3e-14(*NIST*) | 94 |
| $NO_2^-(H_2O)_2 + H_2O + M \rightarrow NO_2^-(H_2O)_3 + M$ | 8.0e-29 | 114 |
| $NO_2^-(H_2O)_3 + M \rightarrow NO_2^-(H_2O)_2 + H_2O + M$ | 1.4e-12(*NIST*) | 95 |
| $NO_2^-(H_2O_2) + M \rightarrow NO_2^- + H_2O_2 + M$ | 1.0e-28(est)/1.7e-12(NIST) | 130 |
| | | |
| $NO_3^- + HFo + M \rightarrow NO_3^-(HFo) + M$ | 1.0e-28(est) | 133 |



| Reaction | Reaction rate coefficient[1] (reference)[2] | Index |
|---|---|---|
| $NO_3^- + H_2O + M \rightarrow NO_3^-(H_2O) + M$ | 1.6e-28 | 115 |
| $NO_3^- + H_2O_2 + M \rightarrow NO_3^-(H_2O_2) + M$ | 1.0e-28(est) | 131 |
| $NO_3^- + NO \rightarrow NO_2^- + NO_2$ | 3.0e-15 | 96 |
| $NO_3^- + O_3 \rightarrow NO_2^- + 2 O_2$ | 1.0e-13 | 97 |
| $NO_3^-(HFo) + M \rightarrow NO_3^- + HFo + M$ | 1.0e-28(est)/9.25e12(NIST) | 134 |
| $NO_3^-(H_2O) + M \rightarrow NO_3^- + H_2O + M$ | 3.0e-14(*NIST*) | 99 |
| $NO_3^-(H_2O) + H_2O + M \rightarrow NO_3^-(H_2O)_2 + M$ | 1.6e-28 | 116 |
| $NO_3^-(H_2O)_2 + M \rightarrow NO_3^-(H_2O) + M$ | 6.81e-13(*Kaz, NIST*) | 100 |
| $NO_3^-(H_2O)_2 + H_2O + M \rightarrow NO_3^-(H_2O)_3 + M$ | 1.6e-28 | 127 |
| $NO_3^-(H_2O)_3 + M \rightarrow NO_3^-(H_2O)_2 + H_2O + M$ | 5.56e-12(NIST) | 128 |
| $NO_3^-(H_2O)_3 + H_2O + M \rightarrow NO_3^-(H_2O)_4 + M$ | 2.0e-29(est) | 196 |
| $NO_3^-(H_2O)_4 + M \rightarrow NO_3^-(H_2O)_3 + H_2O + M$ | 1.0e-11(est, NIST) | 197 |
| $NO_3^-(H_2O_2) + M \rightarrow NO_3^- + H_2O_2 + M$ | 1.0e-28(est)/1.0e10(NIST) | 132 |
| | | |
| $NO_3^{-*} + CO_2 \rightarrow CO_3^- + NO_2$ | 3.0e-9(FF) | 142 |
| $NO_3^{-*} + NO \rightarrow NO_2^- + NO_2$ | 3.0e-9(FF) | 143 |
| $N_2^+ + \sum X^- \xrightarrow{k=6x10^{-8}(300/T)+1.25x10^{-25}[M](300/T)} products$ | 1.2e-8 (Kaz) | 157 |
| | | |
| $O^- + CH_4 \rightarrow HO^- + CH_3$ | 1.e-10 | 59 |
| $O^- + CO_2 + M \rightarrow CO_3^- + M$ | 3.e-28 | 60 |
| $O^- + H_2 \rightarrow HO^- + H$ | 6.e-10 | 62 |
| $O^- + H_2O + M \rightarrow O^-(H_2O) + M$ | 1.3e-28 | 61 |
| $O^- + NO_2 \rightarrow NO_2^- + O$ | 1.25e-9 | 101 |
| $O^- + N_2O \rightarrow NO^- + NO$ | 2.0e-10 | 102 |
| $O^- + O_2 + M \rightarrow O_3^- + M$ | 1.5e-31 | 63 |
| $O^- + O_3 \rightarrow O_3^- + O$ | 8.e-10 | 64 |
| $O^-(H_2O) + H_2O \rightarrow HO^-(H_2O) + HO$ | 6.0e-11(Kaz) | 145 |
| | >1e-11(Alb) | |
| $O^-(H_2O) + O_2 \rightarrow O_3^- + H_2O$ | 6e-11(*Kaz*) | 65 |
| | | |
| $O_2^- + CH_3I \rightarrow I^- + O_2 + CH_3$ | 2.0e-9(*est*) | 4 |
| $O_2^- + CH_3OOH \xrightarrow{M} O_2^-(CH_3OOH)$ | 3e-9(*est*) | 34 |
| $O_2^- + CO_2 + M \rightarrow O_2^-(CO_2) + M$ | 4.7e-29(*FF*) | 5 |
| $O_2^- + H_2O + M \rightarrow O_2^-(H_2O) + M$ | 2.2e-28(FF) | 10 |
| $O_2^- + H_2O_2 \xrightarrow{M} O_2^-(H_2O_2)$ | 3e-9(est) | 26 |
| $O_2^- + NO_2 \rightarrow NO_2^- + O_2$ | 8.0e-10(P) | 103 |
| $O_2^- + O_2 + M \rightarrow O_2^-(O_2) + M$ | 3.8e-30(Kaz) | 55 |
| $O_2^- + O_3 \rightarrow O_3^- + O_2$ | 7.8e-10(Fah) | 8 |
| | 3.0e-10 (*FF*) | |
| | 7.8e-10 (*Fah*) | |
| | 6.0e-10 (*Do*) | |
| | 7.8e-10 (*Kaz*) | |
| | 4.0e-10 (*Pop*) | |
| $O_2^-(CH_3OOH) + M \rightarrow O_2^- + CH_3OOH + M$ | 3e-9/1.1e42(O'Su) | 35 |
| $O_2^-(CH_3OOH) + H_2O \rightarrow O_2^-(H_2O) + CH_3OOH$ | 2e-9/7.7e6(est) | 37 |
| $O_2^-(CO_2) + M \rightarrow O_2^- + CO_2 + M$ | 4.7e-29/2.34e-11(*FF, NIST*) | 6 |



| Reaction | Reaction rate coefficient[1] (reference)[2] | Index |
|---|---|---|
| $O_2^-(CO_2) + CH_3OOH \rightarrow O_2^-(CH_3OOH) + CO_2$ | 2.0e-10 | 161 |
| $O_2^-(CO_2) + CO_2 + M \rightarrow O_2^-(CO_2)_2 + CO_2 + M$ | 1.0e-28(est) | 200 |
| $O_2^-(CO_2) + H_2O \rightarrow O_2^-(H_2O) + CO_2$ | 5.8e-10/2.3(*Alb, FF*) k(T=298)=2.5e-9(*Kaz*) | 13 |
| $O_2^-(CO_2) + H_2O + M \rightarrow O_2^-(CO_2)(H_2O) + M$ | 1e-28 (*est H, P*) | 21 |
| $O_2^-(CO_2) + H_2O_2 \xrightarrow{M} O_2^-(CO_2)(H_2O_2)$ | 2e-11(est) | 22 |
| $O_2^-(CO_2) + H_2O_2 \rightarrow O_2^-(H_2O_2) + CO_2$ | 1.8e-10 | 160 |
| $O_2^-(CO_2) + NO \rightarrow NO_3^{-*} + CO_2 + H_2O$ | 4.8e-11 | 89 |
| $O_2^-(CO_2) + O_3 \rightarrow O_3^- + CO_2 + O_2$ | 7.0e-11(*Alb*) | 46 |
| $O_2^-(CO_2)_2 + M \rightarrow O_2^-(CO_2) + CO_2 + M$ | 3.8e-10(est, NIST) | 201 |
| $O_2^-(CO_2)_2 + H_2O \rightarrow O_2^-(CO_2)(H_2O) + CO_2$ | 1.0e-9(est) | 202 |
| $O_2^-(CO_2)_2 + H_2O_2 \rightarrow O_2^-(CO_2)(H_2O_2) + CO_2$ | 1.0e-9(est) | 203 |
| $O_2^-(CO_2)(H_2O) + M \rightarrow O_2^-(CO_2) + H_2O + M$ | 1.0e28/1.9e-13(*est H, NIST*) | 14 |
| $O_2^-(CO_2)(H_2O) + M \rightarrow O_2^-(H_2O) + CO_2 + M$ | 2.7e-15(est, NIST) | 198 |
| $O_2^-(CO_2)(H_2O) + H_2O \rightarrow O_2^-(H_2O)_2 + CO_2$ | 1.0e-9(Fah, est) | 148 |
| $O_2^-(CO_2)(H_2O) + H_2O + M \rightarrow O_2^-(CO_2)(H_2O)_2 + M$ | 1.0e-28 | 180 |
| $O_2^-(CO_2)(H_2O)_2 + M \rightarrow O_2^-(CO_2)(H_2O) + H_2O + M$ | 1.0e-13 | 181 |
| $O_2^-(CO_2)(H_2O)_2 + H_2O \rightarrow O_2^-(H_2O)_3 + CO_2$ | 1.0e-10 | 182 |
| $O_2^-(CO_2)(H_2O)_2 + O_3 \rightarrow O_3^-(H_2O)_3 + CO_2$ | 1.0e-10 | 183 |
| $O_2^-(CO_2)(H_2O_2) + M \rightarrow O_2^-(CO_2) + H_2O_2 + M$ | 2e-11/2.e12 (*est*) | 23 |
| $O_2^-(CO_2)(H_2O_2) + H_2O \rightarrow O_2^-(CO_2)(H_2O) + H_2O_2$ | 3e-9/7e9(est) 1.0e-14(*est H, M*) | 25 |
| $O_2^-(CO_2)(H_2O) + H_2O_2 \rightarrow O_2^-(CO_2)(H_2O_2) + H_2O$ | 3e-9(est) | 24 |
| $O_2^-(H_2O) + M \rightarrow O_2^- + H_2O + M$ | 2.2e-28/3.04e-11(*FF, NIST*) k(T=298)=4.33e-18(*Kaz*) | 11 |
| $O_2^-(H_2O) + CH_3OOH \rightarrow O_2^-(CH_3OOH) + H_2O$ | 2e-9(est) | 36 |
| $O_2^-(H_2O) + CO_2 \rightarrow O_2^-(CO_2) + H_2O$ | 5.8e-10(*Alb*) k(T=298)=2.5e-9(*Kaz*) | 12 |
| $O_2^-(H_2O) + CO_2 + M \rightarrow O_2^-(CO_2)(H_2O) + M$ | 1.0e-30 | 178 |
| $O_2^-(H_2O) + H_2O + M \rightarrow O_2^-(H_2O)_2 + M$ | 5.4e-28(*PK*) | 50 |
| $O_2^-(H_2O) + H_2O_2 \rightarrow O_2^-(H_2O_2) + H_2O$ | 3e-9(est) | 28 |
| $O_2^-(H_2O) + H_2O_2 + M \rightarrow O_2^-(H_2O_2)(H_2O) + M$ | 1.0e-29(est) | 153 |
| $O_2^-(H_2O) + NO \rightarrow NO_3^- + H_2O$ | 3.1e-10 | 104 |
| $O_2^-(H_2O) + NO_2 \rightarrow NO_2^- + H_2O + O_2$ | 9.0e-10 | 105 |
| $O_2^-(H_2O) + O_2 \rightarrow O_2^-(O_2) + H_2O$ | 2.5e-15 | 82 |
| $O_2^-(H_2O) + O_3 \rightarrow O_3^- + H_2O + O_2$ | 8.0e-10(*Fah*) | 47 |
| $O_2^-(H_2O)_2 + M \rightarrow O_2^-(H_2O) + H_2O + M$ | 1.1e-14(*PK*) | 51 |
| $O_2^-(H_2O)_2 + CH_3OOH \rightarrow\rightarrow O_2^-(CH_3OOH) + 2 H_2O$ | 3.e-11(est) | 151 |
| $O_2^-(H_2O)_2 + CO_2 \rightarrow O_2^-(CO_2)(H_2O) + H_2O$ | 7.0e-11(Fah) | 147 |
| $O_2^-(H_2O)_2 + H_2O + M \rightarrow O_2^-(H_2O)_3 + M$ | 5.e-28 | 70 |
| $O_2^-(H_2O)_2 + H_2O_2 \rightarrow\rightarrow O_2^-(H_2O_2) + 2 H_2O$ | 7.5-10(est) | 149 |
| $O_2^-(H_2O)_2 + NO \rightarrow NO_3^- + 2 H_2O$ | 3.0e-10 | 106 |
| $O_2^-(H_2O)_2 + NO_2 \rightarrow NO_2^- + 2 H_2O + O_2$ | 9.0e-10 | 107 |
| $O_2^-(H_2O)_2 + O_3 \rightarrow O_3^-(H_2O) + H_2O + O_2$ | 7.8e-10(*Fah*) | 52 |
| $O_2^-(H_2O)_3 + M \rightarrow O_2^-(H_2O)_2 + H_2O + M$ | 5.e-28/3.33e-16(*NIST*) | 71 |
| $O_2^-(H_2O)_3 + CH_3OOH \rightarrow\rightarrow O_2^-(CH_3OOH) + 3 H_2O$ | 5.0e-14(est) | 152 |
| $O_2^-(H_2O)_3 + CO_2 \rightarrow O_2^-(CO_2)(H_2O)_2 + H_2O$ | 1.0e-14 | 179 |
| $O_2^-(H_2O)_3 + H_2O + M \rightarrow O_2^-(H_2O)_4 + M$ | 1.0e-28(H) | 192 |

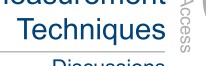



| Reaction | Reaction rate coefficient[1] (reference)[2] | Index |
|---|---|---|
| $O_2^-(H_2O)_3 + H_2O_2 \rightarrow\rightarrow O_2^-(H_2O_2) + 3\ H_2O$ | 1.25e-10(est) | 150 |
| $O_2^-(H_2O)_3 + O_3 \rightarrow O_3^-(H_2O)_2 + H_2O + O_2$ | 6.4e-10(Fah) | 146 |
| $O_2^-(H_2O)_3 + NO \rightarrow NO_3^-(H_2O)_2 + H_2O$ | 1.5e-10(P) | 187 |
| $O_2^-(H_2O)_4 + M \rightarrow O_2^-(H_2O)_3 + H_2O + M$ | 1.2e-12(H, NIST) | 193 |
| $O_2^-(H_2O)_4 + H_2O + M \rightarrow O_2^-(H_2O)_5 + M$ | 5.0e-29(H) | 194 |
| $O_2^-(H_2O)_4 + NO \rightarrow NO_3^-(H_2O)_3 + H_2O$ | 1.2e-10(P) | 188 |
| $O_2^-(H_2O)_5 + M \rightarrow O_2^-(H_2O)_4 + H_2O + M$ | 4.5e-12(H, NIST) | 195 |
| $O_2^-(H_2O)_5 + NO \rightarrow NO_3^-(H_2O)_4 + H_2O$ | 1.2e-10(P) | 189 |
| $O_2^-(H_2O_2) + M \rightarrow O_2^- + H_2O_2 + M$ | 3.e-9/2.96e16(est) | 27 |
| $O_2^-(H_2O_2) + CO_2 + M \rightarrow O_2^-(CO_2)(H_2O_2) + M$ | 3.5e-30(est) | 199 |
| $O_2^-(H_2O_2) + H_2O \rightarrow O_2^-(H_2O) + H_2O_2$ | 3.0e-9/3.9e8(est) | 29 |
| $O_2^-(H_2O_2)(H_2O) + M \rightarrow O_2^-(H_2O_2) + H_2O + M$ | 2.0e-21(est) | 154 |
| $O_2^-(H_2O_2)(H_2O) + M \rightarrow O_2^-(H_2O) + H_2O_2 + M$ | 1.0e-22(est) | 155 |
| $O_2^-(H_2O_2)(H_2O) + CO_2 \rightarrow O_2^-(CO_2)(H_2O_2) + H_2O$ | 7.0e-11(est) | 156 |
| $O_2^-(O_2) + M \rightarrow O_2^- + O_2 + M$ | 1e-14(*Kaz*) | 68 |
| $O_2^-(O_2) + CO_2 \rightarrow O_2^-(CO_2) + O_2$ | 4.3-10(*FF*) | 56 |
| $O_2^-(O_2) + H_2O \rightarrow O_2^-(H_2O) + O_2$ | 1.5e-9(*Ike*) | 57 |
| $O_2^-(O_2) + NO \rightarrow NO_3^{-*} + O_2$ | 2.5e-10 | 108 |
| | | |
| $O_3^- + CH_3OOH + M \rightarrow O_3^-(CH_3OOH) + M$ | 1.0e-28 | 174 |
| $O_3^- + CO_2 \rightarrow CO_3^- + O_2$ | 4.0e-10(*F67*) | 9 |
| $O_3^- + H_2O + M \rightarrow O_3^-(H_2O) + M$ | 1.92e-28(*FF*) | 19 |
| $O_3^- + H_2O_2 + M \rightarrow O_3^-(H_2O_2) + M$ | 1.0e-28 | 166 |
| $O_3^- + NO \rightarrow NO_2^- + O_2$ | 1.1e-12(*50:50*) | 109 |
| $O_3^- + NO \rightarrow NO_3^- + O$ | 1.1e-12(*50:50*) | 110 |
| $O_3^- + NO_2 \rightarrow NO_2^- + O_3$ | 7.0e-10(P) | 191 |
| $O_3^- + NO_2 \rightarrow NO_3^- + O_2$ | 2.8e-10 | 111 |
| | >1e-11(Alb) | |
| $O_3^-(H_2O) + M \rightarrow O_3^- + H_2O + M$ | 1.92e-28/1.46e-15(*FF, NIST*) | 20 |
| $O_3^-(H_2O) + CH_3OOH \rightarrow O_3^-(CH_3OOH) + H_2O$ | 1.0e-9 | 175 |
| $O_3^-(H_2O) + CO_2 \rightarrow CO_3^-(H_2O) + O_2$ | 1.75e10(Do, Wil) | 48 |
| $O_3^-(H_2O) + CO_2 \rightarrow CO_3^- + H_2O + O_2$ | 1.75e10(Do, Wil) | 49 |
| $O_3^-(H_2O) + H_2O + M \rightarrow O_3^-(H_2O)_2 + M$ | 1.92e-28 | 72 |
| $O_3^-(H_2O) + H_2O_2 \rightarrow O_3^-(H_2O_2) + H_2O$ | 1.0e-9 | 167 |
| $O_3^-(H_2O)_2 + M \rightarrow O_3^-(H_2O) + H_2O + M$ | 1.28e-13(*NIST*) | 73 |
| $O_3^-(H_2O)_2 + CH_3OOH \rightarrow O_3^-(CH_3OOH) + 2\ H_2O$ | 1.0e-10 | 176 |
| $O_3^-(H_2O)_2 + CO_2 \rightarrow CO_3^-(H_2O) + H_2O + O_2$ | 1.0e-10(P) | 185 |
| $O_3^-(H_2O)_2 + H_2O + M \rightarrow O_3^-(H2O)_3 + M$ | 1.68e-28 | 74 |
| $O_3^-(H_2O)_2 + H_2O_2 \rightarrow O_3^-(H_2O_2) + 2\ H_2O$ | 1.0e-10 | 168 |
| $O_3^-(H_2O)_3 + M \rightarrow O_3^-(H_2O)_2 + H_2O + M$ | 2.17e-12(*NIST*) | 75 |
| $O_3^-(H_2O)_3 + H_2O_2 \rightarrow O_3^-(H_2O_2) + 3\ H_2O$ | 1.0e-11 | 169 |
| $O_3^-(H_2O)_3 + CH_3OOH \rightarrow O_3^-(CH_3OOH) + 3\ H_2O$ | 1.0e-11 | 177 |
| $O_3^-(H_2O)_3 + CO_2 \rightarrow CO_3^-(H_2O)_2 + H_2O + O_2$ | 5.0e-11(P) | 186 |

[1]Reaction rate coefficient units: unimolecular, $s^{-1}$; bimolecular, $cm^3\ molec^{-1}\ s^{-1}$; termolecular, $cm^6\ molec^2\ s^{-1}$. First listed reaction rate coefficient for a reaction is used in the model mechanism, others are included to show published range of values. A.Ae-n reads as $A.A \times 10^{-n}$.





[2]AK1970 (Arshadi and Kebarle, 1970); Alb (Albritton, 1978); Do1977 (Dotan et al., 1977); F67 (Fehsenfeld et al., 1967); FF1974 (Fehsenfeld and Ferguson, 1974); H (Huertas et al., 1978); HY1992 (Hiraoka and Yamabe, 1992); Ike (Ikezoe et al., 1987) Iyer (Iyer et al., 2016); Kaz (Kazil, 2002); KFP1972 (Kebarle et al., 1972); Kov (Kovacs et al., 2016); M (Mohnen, 1974); NIST (Bartmess, 2016); O'Su (O'Sullivan et al., 2017); P (Popov, 2010); PP1966 (Pack and Phelps, 1966); Wil (Williams et al., 2002).


**Table A3:** Reactions that yield $O_2^-(H_2O_2)$, $O_2^-(CH_3OOH)$, $O_2^-(CO_2)(H_2O_2)$, $I^-(H_2O_2)$, $I^-(HFo)$ and $I^-(HAc)$.

| Cluster Ion | Cluster Ion Source Reaction | Rate | Index |
|---|---|---|---|
| $O_2^-(H_2O_2)$ | | | |
| | $O_2^- + H_2O_2 \xrightarrow{M} O_2^-(H_2O_2)$ | 3.0e-9 | 26 |
| | $O_2^-(H_2O) + H_2O_2 \rightarrow O_2^-(H_2O_2) + H_2O$ | 3.0e-9 | 28 |
| | $O_2^-(H_2O)_2 + H_2O_2 \rightarrow\rightarrow O_2^-(H_2O_2) + 2\ H_2O$ | 7.5-10 | 149 |
| | $O_2^-(H_2O)_3 + H_2O_2 \rightarrow\rightarrow O_2^-(H_2O_2) + 3\ H_2O$ | 1.25-10 | 150 |
| | $O_2^-(H_2O) + H_2O_2 + M \rightarrow O_2^-(H_2O_2)(H_2O) + M$ | 1.0e-29 | 153 |
| | $O_2^-(H_2O_2)(H_2O) + M \rightarrow O_2^-(H_2O_2) + H_2O + M$ | 2.0e-21 | 154 |
| | $O_2^-(CO_2) + H_2O_2 \rightarrow O_2^-(H_2O_2) + CO_2$ | 1.8e-10 | 160 |
| $O_2^-(CH_3OOH)$ | | | |
| | $O_2^- + CH_3OOH \xrightarrow{M} O_2^-(CH_3OOH)$ | 3.0e-9 | 34 |
| | $O_2^-(H_2O) + CH_3OOH \rightarrow O_2^-(CH_3OOH) + H_2O$ | 2.0e-9 | 36 |
| | $O_2^-(H_2O)_2 + CH_3OOH \rightarrow\rightarrow O_2^-(CH_3OOH) + 2\ H_2O$ | 3.0e-11 | 151 |
| | $O_2^-(H_2O)_3 + CH_3OOH \rightarrow\rightarrow O_2^-(CH_3OOH) + 3\ H_2O$ | 5.0e-14 | 152 |
| | $O_2^-(CO_2) + CH_3OOH \rightarrow O_2^-(CH_3OOH) + CO_2$ | 2.0e-10 | 161 |
| $O_2^-(CO_2)(H_2O_2)$ | | | |
| | $O_2^-(CO_2) + H_2O_2 \xrightarrow{M} O_2^-(CO_2)(H_2O_2)$ | 2.e-11 | 22 |
| | $O_2^-(CO_2)(H_2O) + H_2O_2 \rightarrow O_2^-(CO_2)(H_2O_2) + H_2O$ | 3.e-9 | 24 |
| | $O_2^-(H_2O_2)(H_2O) + CO_2 \rightarrow O_2^-(CO_2)(H_2O_2) + H_2O$ | 7.0e-11 | 156 |
| | $O_2^-(H_2O_2) + CO_2 + M \rightarrow O_2^-(CO_2)(H_2O_2) + M$ | 3.5e-30 | 199 |
| | $O_2^-(CO_2)_2 + H_2O_2 \rightarrow O_2^-(CO_2)(H_2O_2) + CO_2$ | 1.0e-9 | 203 |
| $CO_3^-(H_2O_2)$ | | | |
| | $CO_3^- + H_2O_2 + M \rightarrow CO_3^-(H_2O_2) + M$ | 1.0e-28 | 162 |
| | $CO_3^-(H_2O) + H_2O_2 \rightarrow CO_3^-(H_2O_2) + H_2O$ | 1.0e-9 | 163 |
| | $CO_3^-(H_2O)_2 + H_2O_2 \rightarrow\rightarrow CO_3^-(H_2O_2) + 2\ H_2O$ | 1.0e-10 | 164 |
| | $CO_3^-(H_2O)_3 + H_2O_2 \rightarrow\rightarrow CO_3^-(H_2O_2) + 3\ H_2O$ | 1.0e-11 | 165 |
| $CO_3^-(CH_3OOH)$ | | | |
| | $CO_3^- + CH_3OOH + M \rightarrow CO_3^-(CH_3OOH) + M$ | 1.0e-28 | 170 |
| | $CO_3^-(H_2O) + CH_3OOH \rightarrow CO_3^-(CH_3OOH) + H_2O$ | 1.0e-9 | 171 |
| | $CO_3^-(H_2O)_2 + CH_3OOH \rightarrow CO_3^-(CH_3OOH) + 2\ H_2O$ | 1.0e-10 | 172 |
| | $CO_3^-(H_2O)_3 + CH_3OOH \rightarrow CO_3^-(CH_3OOH) + 3\ H_2O$ | 1.0e-11 | 173 |
| $I^-(H_2O_2)$, $I^-(HFo)$ and $I^-(HAc)$. | | | |
| | $I^- + H_2O_2 \xrightarrow{M} I^-(H_2O_2)$ | 1.0e-9 | 30 |
| | $I^-(H_2O) + H_2O_2 \rightarrow I^-(H_2O_2) + H_2O$ | 2.0e-9 | 32 |
| | $I^-(H_2O)_2 + H_2O_2 \rightarrow I^-(H_2O_2) + 2H_2O$ | 2e-9 | 204 |
| | $I^-(H_2O)_3 + H_2O_2 \rightarrow I^-(H_2O_2) + 3H_2O$ | <3e-9 | 205 |
| | $I^- + HFo \xrightarrow{M} I^-(HFo)$ | 1.5e-10 | 38 |
| | $I^-(H_2O) + HFo \rightarrow I^-(HFo) + H_2O$ | 2.0e-9 | 40 |
| | $I^-(H_2O)_2 + HFo \rightarrow I^-(HFo) + 2H_2O$ | 2e-9 | 206 |
| | $I^-(H_2O)_3 + HFo \rightarrow I^-(HFo) + 3H_2O$ | <3e-9 | 207 |
| | $I^- + HAc \xrightarrow{M} I^-(HAc)$ | 7.0e-10 | 42 |
| | $I^-(H_2O) + HAc \rightarrow I^-(HAc) + H_2O$ | 2.0e-9 | 44 |
| | $I^-(H_2O)_2 + HAc \rightarrow I^-(HAc) + 2H_2O$ | 3e-9 | 208 |
| | $I^-(H_2O)_2 + HAc \rightarrow I^-(HAc) + 3H_2O$ | <3e-9 | 209 |

**Table A4:** Reaction sequences leading to potential interferences at m/z 66 [$^{18}O$ of $NO_2^-(H_2O)$, $O_3^-(H_2O)$, and $^{18}O$ of $O_2^-(O_2)$] for $O_2^-(H_2O_2)$ and at m/z 80 [$^{18}O$ of $CO_3^-(H_2O)$ and $NO_3^-(H_2O)$] for $O_2^-(CH_3OOH)$.

| Interference cluster | Source Description | Cluster formation reaction sequence | Index |
|---|---|---|---|
| $^{18}O$ of $NO_2^-(H_2O)$ | from $N_2O$ to $NO^-$ to $NO_2^-$ | $O^- + N_2O \rightarrow NO^- + NO$ | 102 |
| | | $NO^- + NO_2 \rightarrow NO_2^- + NO$ | 118 |
| | from NO to $NO_2^-$ | $CO_3^- + NO \rightarrow NO_2^- + CO_2$ | 83 |
| | | $CO_3^-(H_2O) + NO \rightarrow NO_2^- + CO_2 + H_2O$ | 85 |
| | | $CO_3^-(H_2O) + NO \rightarrow NO_2^-(H_2O) + CO_2$ | 86 |
| | from $NO_2$ to $NO_2^-$ | $HO^- + NO_2 \rightarrow NO_2^- + HO$ | 90 |
| | | $O_2^- + NO_2 \rightarrow NO_2^- + O_2$ | 190 |
| | | $O_3^- + NO_2 \rightarrow NO_2^- + O_3$ | 191 |
| | | $O^- + NO_2 \rightarrow NO_2^- + O$ | 101 |
| | | $O_2^- + NO_2 \rightarrow NO_2^- + O_2$ | 103 |
| | | $O_2^-(H_2O) + NO_2 \rightarrow NO_2^- + H_2O + O_2$ | 105 |
| | | $O_2^-(H_2O)_2 + NO_2 \rightarrow NO_2^- + 2\ H_2O + O_2$ | 107 |
| | | $O_3^- + NO \rightarrow NO_2^- + O_2$ | 109 |
| | then the $NO_2^-$ hydrate | $NO_2^- + H_2O \xrightarrow{M} NO_2^-(H_2O)$ | 112 |
| $O_3^-(H_2O)$ | | $O_2^- + O_3 \rightarrow O_3^- + O_2$ | 8 |
| | | $O_3^- + H_2O \xrightarrow{M} O_3^-(H_2O)$ | 19 |
| $^{18}O$ of $O_2^-(O_2)$ | | $O_2^- + O_2 \xrightarrow{M} O_2^-(O_2)$ | 55 |
| $^{18}O$ of $CO_3^-(H_2O)$ | | $O_2^- + O_3 \rightarrow O_3^- + O_2$ | 8 |
| | | $O_3^- + CO_2 \rightarrow CO_3^- + O_2$ | 9 |
| | | $CO_3^- + H_2O \xrightarrow{M} CO_3^-(H_2O)$ | 15 |
| $NO_3^-(H_2O)$ | from NO to $NO_3^-$ | $O_2^-(H_2O) + NO \rightarrow NO_3^- + H_2O$ | 104 |
| | | $O_2^-(H_2O)_2 + NO \rightarrow NO_3^- + 2\ H_2O$ | 106 |
| | | $O_3^- + NO \rightarrow NO_3^- + O$ | 110 |
| | | $O_2^-(H_2O)_3 + NO \rightarrow NO_3^-(H_2O)_2 + H_2O$ | 187 |
| | | $O_2^-(H_2O)_4 + NO \rightarrow NO_3^-(H_2O)_3 + H_2O$ | 188 |
| | | $O_2^-(H_2O)_5 + NO \rightarrow NO_3^-(H_2O)_4 + H_2O$ | 189 |
| | from $NO_2$ to $NO_3^-$ | $CO_3^- + NO_2 \rightarrow NO_3^- + CO_2$ | 84 |
| | | $CO_3^-(H_2O) + NO_2 \rightarrow NO_3^- + CO_2 + H_2O$ | 87 |
| | | $CO_3^-(H_2O) + NO_2 \rightarrow NO_3^-(H_2O) + CO_2$ | 88 |
| | | $O_3^- + NO_2 \rightarrow NO_3^- + O_2$ | 111 |
| | from $NO_2^-$ to $NO_3^-$ | $NO_2^- + NO_2 \rightarrow NO_3^- + NO$ | 92 |
| | | $NO_2^- + O_3 \rightarrow NO_3^- + O_2$ | 98 |
| | | $NO_2^- + N_2O \rightarrow NO_3^- + N_2$ | 120 |
| | from $HNO_3$ to $NO_3^-$ | $I^- + HNO_3 \rightarrow NO_3^- + HI$ | 91 |
| | then the $NO_3^-$ hydrate | $NO_3^- + H_2O_2 \xrightarrow{M} NO_3^-(H_2O_2)$ | 131 |



**Table A5:** Enthalpy of formation, entropy and Gibb's formation energy for neutral, ion and ion-clusters in the $O_2^- - O_2 - CO_2 - H_2O -$ hydroperoxide system.

| Species or Cluster | $\Delta H_f^o$ kJ mol$^{-1}$ | $S^o$ J $^oK^{-1}$mol$^{-1}$ | $\Delta G_f^o$ kJ mol$^{-1}$ | Reference[1] |
|---|---|---|---|---|
| **Neutral** | | | | |
| $C_{graphite}$ | 0 | 6 | 0 | NIST2016 |
| $CH_3OOH$ | -128 | 281 | -71 | Gold2012 |
| $CO_2$ | -394 | 214 | -394 | NIST2016; Gold2012 |
| $H_2$ | 0 | 131 | 0 | NIST2016; Gold2012 |
| $H_2O$ | -242 | 189 | -229 | NIST2016; Gold2012 |
| $H_2O_2$ | -136 | 233 | -105 | NIST2016; Gold2012 |
| $O_2$ | 0 | 205 | 0 | NIST2016; Gold2012 |
| **Ion or ion-cluster** | | | | |
| $e^-$ | 0 | 23 | 0 | Bartmess [1994] |
| $O_2^-$ | -43 | 210 | -38 | NIST2016; Bartmess [1994] |
| $O_2^-(CH_3OOH)$ | | | **-340**[2] | OSull2017; this work |
| $O_2^-(CO_2)$ | **-516** | **322** | **-481** | this work |
| $O_2^-(CO_2)_2$ | **-937** | **460** | **-880** | this work |
| $O_2^-(CO_2)(H_2O)$ | **-819** | **430** | **-747** | this work |
| $O_2^-(CO_2)(H_2O)_2$ | **-1105** | **555** | **-1000** | this work |
| $O_2^-(CO_2)(H_2O_2)$ | | | **-630** | OSull2017; this work |
| $O_2^-(H_2O)$ | **-362** | **314** | **-318** | this work |
| $O_2^-(H_2O)_2$ | **-676** | **398** | **-588** | this work |
| $O_2^-(H_2O)_3$ | **-982** | **469** | **-846** | this work |
| $O_2^-(H_2O)_4$ | | | **-1092** | this work |
| $O_2^-(H_2O)_5$ | | | **-1335** | this work |
| $O_2^-(H_2O_2)$ | | | **-237** | OSull2017; this work |
| $O_2^-(O_2)$ | **-115** | **310** | **-79** | NIST2016; AK1970; this work |

[1]AK1970 [Arshadi and Kebarle, 1970]; Gold2012 [Goldsmith et al., 2012]; NIST2016 [Bartmess, 2016]; OSull2017 [O'Sullivan et al., 2017].

5  [2]Note: **underline bold** indicates value derived in this work using published formation and reaction enthalpy, entropy and Gibb's energy.





**Table A6:** Reaction enthalpy, entropy and Gibb's energy for neutral, ion and ion-cluster reactions in the $O_2^- - O_2 - CO_2 - H_2O - $ hydroperoxide system.

| Reaction | $\Delta H_r^o$ kJ mol$^{-1}$ | $\Delta S_r^o$ J °K$^{-1}$mol$^{-1}$ | $\Delta G_r^o$ kJ mol$^{-1}$ | Reference[1] |
|---|---|---|---|---|
| $O_2^- + CO_2 \rightleftharpoons O_2^-(CO_2)$ | -80 | -101 | -49 | NIST2016, HY1992 |
| $O_2^-(CO_2) + CO_2 \rightleftharpoons O_2^-(CO_2)_2$ | -28 | -76 | -5 | NIST2016, HY1992 |
| $O_2^-(CO_2) + H_2O \rightleftharpoons O_2^-(H_2O) + CO_2$ | **2**[2] | **17** | **-3** | NIST2016, FF1974 ($\Delta G_r^o = -2.1$), this work |
| $O_2^-(CO_2) + H_2O \rightleftharpoons O_2^-(CO_2)(H_2O)$ | -61 | **-81** | **-37** | KFP1972, this work |
| $O_2^-(CO_2)(H_2O) + H_2O \rightleftharpoons O_2^-(H_2O)_2 + CO_2$ | **-9** | **-7** | -7 | NIST2016, FF1974, KFP1972, this work |
| $O_2^-(CO_2)(H_2O) + H_2O \rightleftharpoons O_2^-(CO_2)(H_2O)_2$ | -44 | **-65** | -25 | Mohnen [1972], HFG1978, this work |
| $O_2^- + H_2O \rightleftharpoons O_2^-(H_2O)$ | -77 | -84 | -52 | NIST2016, AK1970 |
| $O_2^-(H_2O) + H_2O \rightleftharpoons O_2^-(H_2O)_2$ | -72 | -105 | -41 | NIST2016, AK1970 |
| $O_2^-(H_2O) + CO_2 \rightleftharpoons O_2^-(CO_2)(H_2O)$ | **-63** | **-98** | -34 | NIST2016, FF1974, this work |
| $O_2^-(H_2O)_2 + CO_2 \rightleftharpoons O_2^-(CO_2)(H_2O) + H_2O$ | **9** | **7** | **7** | this work |
| $O_2^-(H_2O)_2 + H_2O \rightleftharpoons O_2^-(H_2O)_3$ | -64 | -118 | -29 | NIST2016, AK1970 |
| $O_2^-(H_2O)_3 + H_2O \rightleftharpoons O_2^-(H_2O)_4$ | | | -18 | NIST2016, AK1970 |
| $O_2^-(H_2O)_4 + H_2O \rightleftharpoons O_2^-(H_2O)_5$ | | | -14 | NIST2016, AK1970 |
| $O_2^- + O_2 \rightleftharpoons O_2^-(O_2)$ | -44 | -102 | -13 | NIST2016 |
| $O_2^- + CH_3OOH \rightleftharpoons O_2^-(CH_3OOH)$ | | | **-231** | OSull2017, this work |
| $O_2^- + H_2O_2 \rightleftharpoons O_2^-(H_2O_2)$ | | | **-99** | OSull2017, this work (-94) |
| $O_2^-(CO_2) + H_2O_2 \rightleftharpoons O_2^-(CO_2)(H_2O_2)$ | | | -43 | OSull2017 |
| $O_2^-(H_2O) + CH_3OOH \rightleftharpoons O_2^-(CH_3OOH) + H_2O$ | | | **-181** | OSull2017, this work |
| $O_2^-(H_2O) + H_2O_2 \rightleftharpoons O_2^-(H_2O_2) + H_2O$ | | | **-42** | this work |
| $O_2^-(H_2O)_2 + H_2O_2 \rightleftharpoons O_2^-(H_2O_2) + 2H_2O$ | | | **-1** | this work |
| $O_2^-(H_2O)_3 + H_2O_2 \rightleftharpoons O_2^-(H_2O_2) + 3H_2O$ | | | **28** | this work |
| $O_2^-(H_2O_2) + CO_2 \rightleftharpoons O_2^-(CO_2)(H_2O_2)$ | | | **7** | this work |

[1]AK1970 [Arshadi and Kebarle, 1970]; FF1974 [Fehsenfeld and Ferguson, 1974]; HFG1978 [Huertas et al., 1978]; HY1992 [Hiraoka and Yamabe, 1992]; KFP1972 [Kebarle et al., 1972]; NIST2016 [Bartmess, 2016]; OSull [O'Sullivan et al., 2017].

[2]Note: **underline bold** indicates value derived in this work using published formation and reaction enthalpy, entropy and Gibb's energy.