# Peer review of "An ion-neutral model to investigate chemical ionization mass spectrometry analysis of atmospheric molecules - application to a mixed reagent ion system for hydroperoxides and organic acids"

_Atmospheric Measurement Techniques, 2017_

## Referee Comment (RC1) · Anonymous Referee #1 · 22 Sep 2017

Heikes et al. have developed a model including some hundreds of ion - molecule reactions, and fit key parameters to match observed pressure and humidity dependencies of the sensitivities for four analyte molecules (formic acid, acetic acid, H2O2 and CH3OOH) measured with a mixed reagent ion (mainly O2- and I-) - CIMS instrument. As a specific application, the authors show how the model can be used to monitor (and presumably correct for) interferences caused by isobaric cluster ion species. The work is interesting and valuable to the atmospheric science community, and I would encourage the authors to upload their model online in a user-friendly format in addition

to reporting it in this article. (I strongly suspect that many more applications will be found for such a model in addition to the ones presented in the article). The authors have gone to great lengths in collecting and "homogenizing" a large amount of data from different sources, some of which are not easily rendered into compatible format. For example, ab initio calculations are typically good at computing relative energetics (e.g. reaction energies), but often fail at reproducing absolute energies (e.g. standard formation free energies, particularly for radical systems, and especially for complex radical clusters). Similarly (as the authors note), experiments performed for example with different carrier gases may yield quite different termolecular rate coefficients. The discrepancies discussed in the supplemental information are thus not particularly surprising. However, the authors seem well aware of these issues, and as far as I can tell have made reasonable and justified choices in how to assemble their model from the fragmented and sometimes contradictory source data they had available.

I warmly recommend that this manuscript be publised in AMTD. I have a few minor questions, comments and further suggestions, which I hope the authors will consider when writing the final version of the manuscript.

-As a general comment, could the authors use their model (and/or the chemical understanding they have gained while constructing it) to qualitatively explain to a nonexpert reader the reason for the "parabolic" behaviour exhibited by the sensitivity of so many analyte/ion combinations with respect to humidity? (E.g. figures 5,6,7 in their manuscript, and many cases of such behaviour reported elsewhere). Why does the sensitivity in these cases first increase, and then decrease, with increasing humidity? I have always assumed the initial increase is due to energy (non)accommodation effects, i.e. water stabilizing the ion-molecule clusters - I guess this is included in the model by reactions of the type R2 being termolecular, while R3 is bimolecular. Is this the explanation for the initial increase, or is the explanation more subtle? And is the subsequent decrease then related mainly to competition by polyhydrates, or what is going on? Also, why does I-(HFo) go up with H2O while I-(HAc) goes down? The
model should presumably provide some insight into this. A separate section explaning the reasons for the observed trends (in the SI if the authors feel the manuscript is too long) would be very useful to readers trying to understand the chemistry behind the CIMS measurements.

-Line 79: the word "expected" is a bit confusing: why did the authors "expect" to see these signals if the reactions forming the hydrates in question have too positive enthalpies/free energies? Am I missing something here? (The whole sentence is actually a bit hard to parse, I would suggest reformulating it.)

-The resolution of the figures are quite poor, making the figures hard to read - could you please make higher-quality figures? (Perhaps the poor resolution is due to some format conversion issue?) Also some of the figures could be made larger (e.g. Fig 1 & 2).

-I found the concept that the acidity of XH (or alternatively basicity of X-) can be used to quantitatively predict the (relative) binding of X- to different analytes quite interesting (R5 and surrounding discussion). Could this correlation be used more generally?

-Line 220-221, please specify that the pictures are plotted for the case A=H2O2 (this is sort of implicit from the discussion above, but specifying it would help the reader).

-Line 375, "too peaked" is a bit ambiguous, please reformulate

-Line 394: "preoxide" => "peroxide"

-Line 98 of the Supplemental Information: the number  $10^{-2} \text{ J} / \text{mol K}$  seems too small (comparing e.g. to entropies given in the tables), pehaps the authors meant kJ / mol K, or alternatively  $10^{+2}$ ?

---

## Referee Comment (RC2) · Anonymous Referee #2 · 22 Dec 2017

This paper aims to develop and test a model of ion-neutral chemistry occurring in a reduced pressure chemical ionization flow reactor whereby ions are generated using a mixture of N2, O2, and CH3I to produce O2- and I- simultaneously.

The model focuses mostly on reactions stemming from clusters of the main reagent ions with water, and the implications for detection of formic acid, acetic acid, H2O2, and CH3OOH product ions.

The paper is well written, and the methodology largely sound, if a bit incomplete (see

below). I don't have any scientific or technical concerns that would prevent this paper from being published. The paper can be accepted as is, but the authors may wish to address a few minor comments listed below.

1) The authors have done a commendable job trying to develop a mechanistic understanding of the ion chemistry occurring in their chemical ionization region. The ion chemistry chosen seems rather complicated, but it seems the broad trends can be rationalized. One aspect I didn't see directly addressed was proton abstraction from weak acids by $O_2^-$. Could this be more explicitly stated as to whether e.g. acetate ions or other carboxylate ions are produced, and if so what would be the resulting secondary ion chemistry in the flow reactor. Would that produce other possible interfering ions?

2) It is not clear how $O_3^-$ and $CO_3^-$ ions are generated.

3) The role of $O_3$ was not well developed in that I couldn't follow why it was tested aside from possibly causing an interfering ion in the $O_2^-$ chemistry. What about $I^-$ chemistry?

4) There are some rate constants given without units.

---

## Author Comment (AC1) · 19 Jan 2018

See supplement which contains author response to reviewer #1 comments, author response to reviewer #2 comments, the revised manuscript showing changes, and the revised supplemental information showing changes.

Please also note the supplement to this comment:
https://www.atmos-meas-tech-discuss.net/amt-2017-286/amt-2017-286-AC1-supplement.pdf

---

## Author Response (AR1)

Response to Anonymous Reviewer #1

The author's responses to anonymous reviewer #1's comments are found after each comment. Each comment is in italics and our response is in a normal font. Yellow highlighting is used to identify changes made to the manuscript. We have removed the reviewer's synopsis and recommendation found at the top of the review.

*Anonymous Referee #1*

*-As a general comment, could the authors use their model (and/or the chemical understanding they have gained while constructing it) to qualitatively explain to a nonexpert reader the reason for the "parabolic" behaviour exhibited by the sensitivity of so many analyte/ion combinations with respect to humidity? (E.g. figures 5,6,7 in their manuscript, and many cases of such behaviour reported elsewhere). I have always assumed the initial increase is due to energy (non)accommodation effects, i.e. water stabilizing the ion-molecule clusters - I guess this is included in the model by reactions of the type R2 being termolecular, while R3 is bimolecular. Is this the explanation for the initial increase, or is the explanation more subtle? And is the subsequent decrease then related mainly to competition by polyhydrates, or what is going on? Also, why does I-(HFo) go up with H2O while I-(HAc) goes down? The model should presumably provide some insight into this. A separate section explaining the reasons for the observed trends (in the SI if the authors feel the manuscript is too long) would be very useful to readers trying to understand the chemistry behind the CIMS measurements.*

The reviewer has hit upon the very reason the model was developed. The variability in sensitivity observed with changing water vapor mixing ratio and sample pressure depended upon the reagent ion-analyte pair. To bring this to the fore we have added the following text to the introduction (approximately new line 59).

As will be shown below within the discussion section, depending upon the reagent ion-analyte pair, the effect of water vapor can lead to:
1) a relative constant sensitivity as water vapor mixing ratios increase until near maximum water vapor mixing ratios are encountered after which the sensitivity decreases with increasing water vapor,
2) sensitivity increases as water vapor mixing ratio increases,
3) sensitivity decreases as water vapor mixing ratio increases, and
4) "parabolic" response in which the sensitivity to an ion-analyte cluster is low at low water vapor mixing ratio passes through a maximum sensitivity at an intermediate water vapor mixing ratio and is low again at high water vapor mixing ratio
and ambient pressure changes can lead to decreasing or increasing sensitivity with an increase in sample pressure (flow).

Further the discussion of the schematic mechanism outlined by R1-R4 was intended to explain this behavior. To improve on our explanation there, we have included sets of conditions or assumptions which lead to the observed sensitivity behavior (approx. line 220).

This simple system with a variety of reaction rate coefficients can yield a suite of sensitivity responses to water vapor.
Case 1: Sensitivity independent of water.
   Assumptions, water vapor does not deplete the reagent ion concentration and the product $k_{R2f}[M][X]$ is much larger than $k_{R3f}K_{R1}[H_2O]$ over the range of $[H_2O]$ encountered.
Case 2: Sensitivity increases with water vapor mixing ratio.
   Assumptions, water vapor does not deplete the reagent ion concentration and the product $k_{R2f}[M][X]$ is smaller than $k_{R3f}K_{R1}[X][H_2O]$ over the range of $[H_2O]$ encountered.
Case 3: Sensitivity decreases with water vapor mixing ratio.
   Assumption set A: water vapor depletes the reagent ion concentration via R1 and
      successive hydration reactions represented by:
   $$X^-(H_2O)_{m-1} + H_2O + M \leftrightarrow X^-(H_2O)_m + M$$
      (R1')
      and reactions like R3f are slow.
   Assumption set B: product $k_{R3r}[X\text{-}(A)][H_2O]$ becomes progressively larger than the sum of $k_{R2f}[M][X\text{-}]$ and $k_{R3f}K_{R1}[X][H_2O]$ as $[H_2O]$ increases.

*-Line 79: the word "expected" is a bit confusing: why did the authors "expect" to see these signals if the reactions forming the hydrates in question have too positive enthalpies/free energies? Am I missing something here? (The whole sentence is actually a bit hard to parse, I would suggest reformulating it.)*

Agree with the reviewer. The intent was to indicate the effective "declustering energy" of the CDC (collision dissociation chamber, entrance plate and octopole). This section has been rewritten for clarity (approximately new line 76).

The physical PCIMS instrument is described in O'Sullivan et al. (2017), Treadaway (2015) and Treadaway et al. (2017). A physical description of the instrument and calibration schemes are presented in the Appendix. The instrument flow and electronic configuration described in the Appendix was used throughout the field and laboratory work reported here. The PCIMS m/z range was 1-500 m/z and the mass resolution was 1.0 m/z. The main component effecting ion-cluster transmission through the system was the collision dissociation chamber (CDC) consisting of an entrance plate and octopole ion guide. The CDC plate DC voltage and the octopole DC and RF voltages were adjusted to maximize the transmission of the hydroperoxide analyte cluster ions $O_2^-(CO_2)(H_2O_2)$ and $O_2^-(CH_3OOH)$ and to reduce the signal from other ions near their respective masses. To estimate the "declustering" energy employed, an analysis of the thermodynamics of the following hydration reactions:
$$O_2^-(H_2O)_{n-1} + H_2O \rightleftarrows O_2^-(H_2O)_n; \quad n = 1-5,$$

$$CO_3^-(H_2O)_{n-1} + H_2O \rightleftarrows CO_3^-(H_2O)_n; \quad n = 1 - 3,$$
$$NO_3^-(H_2O)_{n-1} + H_2O \rightleftarrows NO_3^-(H_2O)_n; \quad n = 1 - 2,$$

and

$$I^-(H_2O)_{n-1} + H_2O \rightleftarrows I^-(H_2O)_n; \quad n = 1 - 4,$$

and the PCIMS signals of the respective hydrates was done. There was an absence of signals for $O_2^-(H_2O)_{n>3}$, $CO_3^-(H_2O)_{n>2}$ and $I^-(H_2O)_{n>1}$ clusters. The absence of these clusters suggested a CDC "declustering" enthalpy cut off at -50 kJ mol$^{-1}$ or a CDC "declustering" Gibbs energy of -20 kJ mol$^{-1}$. The thermodynamic data used in this analysis were from NIST Chemistry WebBook SRD69 (Bartmess, 2016).

*-The resolution of the figures are quite poor, making the figures hard to read - could you please make higher-quality figures? (Perhaps the poor resolution is due to some format conversion issue?) Also some of the figures could be made larger (e.g. Fig 1 & 2).*

Agree with the reviewer. Figures have been redrafted to improve quality and readability for publication in AMT.

*-I found the concept that the acidity of XH (or alternatively basicity of X-) can be used to quantitatively predict the (relative) binding of X- to different analytes quite interesting (R5 and surrounding discussion). Could this correlation be used more generally?*

The short answer is yes and besides the reference to Böhringer et al. [1984] a few examples are given here. The first use of gas phase acidity we have found as it relates to our work comes from Dzidic et al. [1974] in which he discusses oxygen superoxide ions, the transfer of a proton from several organic compounds, specific to our interests, from acetic acid with the production of an acetate ion. Gas phase acidity was more recently used by Veres et al. [2008] and Roberts et al. [2010] in an analysis of negative ion proton transfer chemical ionization as it relates to the use of acetate ion as a reagent to ionize organic acids with lower acidities. Bertram et al. [2011] used gas phase acidity in a discussion of two possible outcomes of acetate chemistry proton transfer resulting in both negative analyte ion formation by proton transfer or in acetate-analyte cluster formation. Bertram [Pers. Comm., Department of Chemistry, University of Wisconsin, Madison, WI) further explored the use of oxygen superoxide ion (O2-) as a reagent ion for formic acid (HFo) and observed both O2-(HFo) clusters and Fo- as products (approximately at a 3:1 proportion).

Oxygen superoxide ion reactions with organic compounds were specifically raised by Anonymous Reviewer 2 and its chemistry is commented on in our response to Anonymous Reviewer 2.

*-Line 220-221, please specify that the pictures are plotted for the case A=H2O2 (this is sort of implicit from the discussion above, but specifying it would help the reader).*

We have added the following sentence to the text at line 259

Fig 2 is plotted with $A = H_2O_2$ as introduced in R2-R4.
and the following sentence to the figure caption for Fig. 2.
R1-R4 refer to reactions introduced in the text with $A = H_2O_2$.

*-Line 375, "too peaked" is a bit ambiguous, please reformulate*

This section now near line 417-421 was revised and now reads:
The simulated $O_2^-$ hydroperoxide cluster sensitivity showed too steep of an increase to the maximum value and then too steep of a decrease after the maximum as water vapor mixing ratio varied from 10 to 20,000 ppm. The maximum sensitivity was at water vapor mixing ratios near a few 100 ppm.  The addition of reactions leading to higher order ion hydrates, $(H_2O)_{n>1}$, carbonates, $(CO_2)_{n>1}$, mixed hydrate-carbonates, $(H_2O)_m(CO_2)_n$, were included to reduce the steepness on each side of the maximum (see supplemental information for details).

*-Line 394: "preoxide" => "peroxide"*

This was corrected and is at new line 436.

*-Line 98 of the Supplemental Information: the number 10^-2 J / mol K seems too small (comparing e.g. to entropies given in the tables), pehaps the authors meant kJ / mol K, or alternatively 10^+2?*

The reviewer is correct, the intended value was of order $10^2$ J mol$^{-1}$ K$^{-1}$ and the text in line 98 has been revised.

Response to Anonymous Reviewer #2

The author's responses to anonymous reviewer #1's comments are found after each comment. Each comment is in *italics* and our response is in a normal font. Cyan highlighting is used to identify changes made to the manuscript. We have removed the reviewer's synopsis and recommendation found at the top of the review.

*Anonymous Referee #2*

1) *The authors have done a commendable job trying to develop a mechanistic understanding of the ion chemistry occurring in their chemical ionization region. The ion chemistry chosen seems rather complicated, but it seems the broad trends can be rationalized. One aspect I didn't see directly addressed was proton abstraction from weak acids by O2-. Could this be more explicitly stated as to whether e.g. acetate ions or other carboxylate ions are produced, and if so what would be the resulting secondary ion chemistry in the flow reactor. Would that produce other possible interfering ions?*

The reviewer is correct and we did not include or discuss reactions involving proton abstraction by oxygen superoxide ions, $O_2^-$. We did consider this at a cursory level because of the work of Dzidic et al. [J. Amer. Chem. Soc., 96, 5258-5259, 1974] and the summary of reaction rate constants between $O_2^-$ and organic compounds by Ikezoe et al. [1987]. We believed the paucity of reaction rate coefficients and the lack of organic compound measurements in our laboratory experiments would make model predictions extremely speculative. At least with our sensitivity work, we could constrain the model with observations.

The reviewers comment did encourage us to reevaluate this process in the context of our modeling activity. We note the use of gas phase acidity to evaluate acetate ion as a reagent ion in negative-ion proton-transfer by Veres et al. [Int'l J. Mass Spect. 274, 48–55, 2008], Roberts et al. [Atmos. Meas. Tech., 3, 981-990, 2010] and Bertram et al. [Atmos. Meas. Tech., 4, 1471–1479, 2011]. In Bertram et al., they point out that acetate ion chemistry proceeds through two channels, proton transfer as the reviewer noted and by cluster ion formation:
$$CH_3C(O)O^- + HAnalyte \rightarrow CH_3C(O)OH + Analyte^-$$
$$CH_3C(O)O^- + HAnalyte \rightarrow CH_3C(O)O^-(HAnalyte)$$
Further, Bertram [Timothy Bertram, Pers. Comm., Department of Chemistry, University of Wisconsin, Madison, WI, 2018] shared observations from his laboratory in which they worked with $O_2^-$ as a potential CIMS reagent ion for formic acid. He observed both formate ion (proton abstraction channel) and the $O_2^-(HC(O)OH)$ cluster ion as products at approximately a 1:3 proportion, respectively. In our own laboratory work, Treadaway et al. [Atmos. Meas. Tech. Disc., 2017, doi:10.5194/amt-2017-344] included results showing $O_2^-$ cluster ion formation with acetic acid and hydroxyacetaldehyde. She reviewed the limited data in hand and within the precision of the CIMS quadrupole instrument did not detect acetate ion or the $CH(O)CH_2O^-$ (hydroxy acetaldehyde anion) at m/z 59. As a significant caveat, 59 is adjacent to $CO_3^-$ at m/z 60 which is a major ion in our system.

We have added the following in the discussion section to address the reviewers general concern regarding potential interferences and their specific comment regarding acetate. (near line 586).

In review, it was pointed out the model did not include the ion-neutral chemistry of organic compounds, specifically the potential for oxygen superoxide ion to abstract a proton from acetic acid and subsequently acetate ion to abstract a proton from weaker acids. Currently, the paucity of reaction rate coefficients for $O_2^- + organic$ reactions precludes their inclusion in our model. We must note, the potential exists for such chemistry to impact the simulations and we are unable to quantitatively assess their importance.

*2) It is not clear how O3- and CO3- ions are generated.*

The reaction pathways leading to O3- and CO3- ions are explicitly presented in Table A4 as they relate to potential interferences. This was implicit in the discussion of interferences now at lines 549 and 550, which we have modified to be more explicit and now reads:

The interference ion production pathways involving $NO_2^-$, $NO_3^-$, $CO_3^-$, $O_3^-$, and $O_2^-(O_2)$ are outlined in the Appendix, Table A4.

*3) The role of O3 was not well developed in that I couldn't follow why it was tested aside from possibly causing an interfering ion in the O2- chemistry. What about I- chemistry?*

Gas phase chemistry of I- with O3 proved difficult to find. I-O3 cluster formation and loss is included (Table A2). The bimolecular reaction rate coefficient of cluster formation is about a factor of 100 smaller than the collision limit. The rate data are from Williams et al. [2002]. This same reference states: *"The I- ion was observed to cluster with O3 to form IO3- with a rate constant of approximately 1x10$^{-11}$ cm$^3$ s$^{-1}$, which is a factor of 2 above our detection limit, and no other product channels were observed."* This reference further goes on to state the rates of reaction of I- with O2 is less than 1x10$^{-13}$ cm$^3$ s$^{-1}$. Consequently, we did not explore this path further with respect to ozone or oxygen.

*4) There are some rate constants given without units.*
Reaction rate coefficient units are specified in Table A2 and in the text (new lines 172-173)

[revised manuscript text omitted]